# Zero-shot HOI Detection with MLLM-based Detector-agnostic Interaction Recognition

**Shiyu Xuan**[1], **Dongkai Wang**[2], **Zechao Li**[1,*]**Jinhui Tang**[3]

[1]School of Computer Science and Engineering, Nanjing University of Science and Technology
[2]School of Computing and Artificial Intelligence, Southwestern University of Finance and Economics
[3]Nanjing Forestry University
shiyu_xuan@njust.edu.cn, wdk@swufe.edu.cn, zechao.li@njust.edu.cn, tangjh@njfu.edu.cn

## Abstract

Zero-shot Human-object interaction (HOI) detection aims to locate humans and objects in images and recognize their interactions. While advances in open-vocabulary object detection provide promising solutions for object localization, interaction recognition (IR) remains challenging due to the combinatorial diversity of interactions. Existing methods, including two-stage methods, tightly couple IR with a specific detector and rely on coarse-grained vision-language model (VLM) features, which limit generalization to unseen interactions. In this work, we propose a decoupled framework that separates object detection from IR and leverages multi-modal large language models (MLLMs) for zero-shot IR. We introduce a deterministic generation method that formulates IR as a visual question answering task and enforces deterministic outputs, enabling training-free zero-shot IR. To further enhance performance and efficiency by fine-tuning the model, we design a spatial-aware pooling module that integrates appearance and pairwise spatial cues, and a one-pass deterministic matching method that predicts all candidate interactions in a single forward pass. Extensive experiments on HICO-DET and V-COCO demonstrate that our method achieves superior zero-shot performance, strong cross-dataset generalization, and the flexibility to integrate with any object detectors without retraining. The codes are publicly available at https://github.com/SY-Xuan/DA-HOI.

## 1 Introduction

Human-object interaction (HOI) detection aims to localize humans and objects in an image and recognize the interactions between them. It provides a fine-grained understanding of human activities, which is crucial for downstream applications such as robotic manipulation (Shridhar et al., 2022; Jin et al., 2024), image captioning (Wu et al., 2022; Yao et al., 2018), and autonomous driving (Liao et al., 2025; Wang et al., 2023). Despite recent progress, HOI detection remains challenging due to the large combinatorial space of human-object pairs and the need for fine-grained visual understanding. Moreover, real-world applications often require recognizing unseen human-object interactions that do not appear in the training set, leading to the task of *zero-shot HOI detection*.

To address zero-shot HOI detection, recent works (Ning et al., 2023; Lei et al., 2025; 2024b; 2023) exploit vision-language models such as CLIP (Radford et al., 2021) to transfer semantic knowledge from language to HOI detection. Typically, they use text embeddings of interaction descriptions, *e.g.*, "A person is holding a cup", to construct classifiers for unseen interactions. While effective to some extent, these methods face limitations. First, CLIP features lack the fine-grained representational capacity required to distinguish visually similar interactions. Methods have to incorporate detector features to obtain fine-grained representation of instances. Second, their core mechanism, aligning visual and text features, operates only on categories observed during training, which restricts generalization to unseen interactions. Progress in open-vocabulary object detection (Liu et al., 2024; Cheng et al., 2024) provides a sufficient solution for localizing unseen objects, while interaction recognition (IR) remains the main bottleneck. As shown in Fig. 1 (a), most existing methods,

---

*Corresponding Author

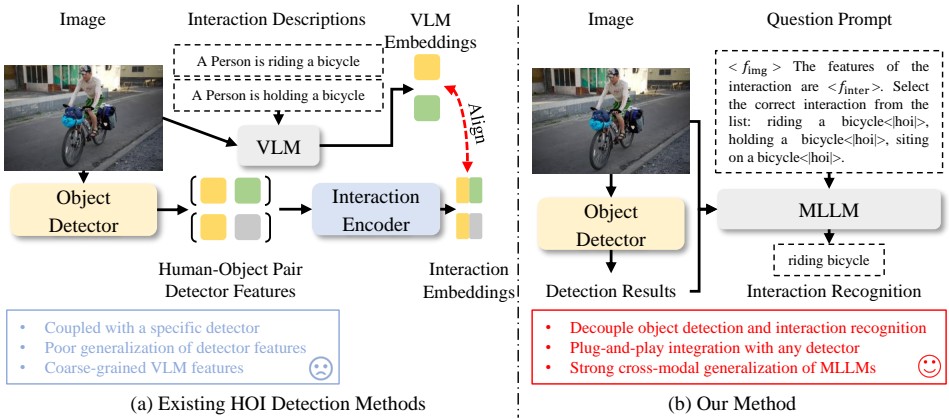

Figure 1: (a) Existing methods, including two-stage methods, couple object detection and interaction recognition together. Their performance are constrained by the limited generalization of detector features and coarse-grained VLM features. (b) Our method fully decouples these two processes and harnesses the powerful MLLMs for interaction recognition. This design benefits from the generalization of MLLMs and advanced detectors.

including two-stage methods (Mao et al., 2024; Lei et al., 2025; Kim et al., 2025), couple IR with a specific object detector (Carion et al., 2020). These methods rely on detector features or inter-object relations to enhance VLM features with fine-grained representational capacity, but struggle to balance fine-grained detector features with generalized VLM features. As a result, they limit independent improvements to IR and detection, and the detector cannot be changed without retraining.

To overcome these limitations, we propose a decoupled framework that separates object detection from IR. Effective IR requires representations that are both fine-grained and broadly generalizable. Unlike CLIP-based methods that rely on static embeddings, multi-modal large language models (MLLMs) are trained on large-scale image-text pairs and instruction-following tasks, equipping them with strong cross-modal generalization. This motivates us to exploit MLLMs for IR in zero-shot HOI detection. By decoupling object detection and IR, our framework enables plug-and-play integration with any object detector, allowing us to focus on advancing IR without being constrained by the training, architecture, and limited generalization capacity of the detector.

Building on this foundation, for each human–object pair, we formulate IR as a visual question answering (VQA) task, where both the pair information and candidate interactions are encoded into prompts for the MLLM. To adapt the open-ended text generation of MLLMs to the multi-label classification setting of IR, we introduce a deterministic generation method that restricts the answer within the candidate interaction list, establishing a strong training-free IR method.

Despite these improvements, two challenges remain. First, the features extracted from human and object bounding boxes are sensitive to imperfect detections and fail to capture the pairwise spatial information crucial for IR. Second, deterministic generation requires multiple forward passes for IR, resulting in significant computational overhead when the candidate list is large. To address these issues, we propose a spatial-aware pooling module that incorporates both appearance and pairwise spatial cues, and a one-pass deterministic matching strategy that reformulates generation as feature matching, allowing all candidate interactions to be predicted within a single forward pass. Fine-tuned on a training set, these components further enhance both performance and efficiency.

We evaluate our method on HICO-DET and V-COCO, demonstrating superior performance compared to existing zero-shot methods, while also validating strong cross-dataset generalization, *e.g.*, outperforming CMMP by 12.26%. This is an early work to fully decouple the object detection and IR in HOI detection. By integrating the MLLM for IR, our deterministic generation method enables training-free IR, while the proposed spatial-aware pooling and one-pass deterministic matching further improve fine-tuning performance and inference efficiency. Our framework achieves promising results in both zero-shot and cross-dataset settings. Once trained, our method has the flexibility to integrate with any object detector without retraining, establishing a new paradigm for HOI detection with MLLMs.

## 2 RELATED WORKS

**Human-Object Interaction Detection.** HOI detection methods are generally divided into two categories: two-stage and one-stage. Two-stage methods follow a detection-then-recognition pipeline, where humans and objects are first localized and then paired for interaction recognition. Early works such as InteractNet (Gkioxari et al., 2018) and iCAN (Gao et al., 2018) enhance appearance modeling through human-centric or attention-based features. Beyond appearance cues, some methods integrate spatial cues and structural modeling via GNNs or attention mechanisms (Gao et al., 2020; Qi et al., 2018; Zhang et al., 2022a; Ulutan et al., 2020; Zhang et al., 2021; 2023), while others exploit human pose or part-level features (Moon et al., 2021; Wan et al., 2019; Zhou & Chi, 2019). One-stage methods directly predict human–object–interaction triplets. Point-based methods (Liao et al., 2020; Wang et al., 2020) reformulate pairs as keypoints, while query-based methods (Kim et al., 2021; Zhou et al., 2022; Zou et al., 2021; Chen et al., 2021; Tamura et al., 2021) leverage query-based decoding to predict triplets end-to-end. Follow-up works further improve performance through query optimization (Dong et al., 2021; 2022), large-scale pre-training (Li et al., 2024b), and structural enhancements (Zhang et al., 2022b; Kim et al., 2022; Tu et al., 2022).

**Zero-shot HOI Detection.** Conventional methods fail to generalize to unseen interactions. Early attempts (Hou et al., 2020; 2021) disentangle object and verb features for compositional generalization, but cannot handle unseen objects or verbs. With the emergence of vision–language models (VLMs) such as CLIP (Radford et al., 2021), recent methods exploit text embeddings of interaction descriptions for semantic transfer. For instance, Gen-VLKT (Liao et al., 2022) aligns interaction features with CLIP embeddings, while HOICLIP (Ning et al., 2023) enhances fine-grained recognition via verb adapters. ADA-CM (Lei et al., 2023) introduces concept-guided memory for training-free or fine-tuning scenarios. LAIN (Kim et al., 2025) improves locality and interaction awareness.

**Leveraging MLLMs for HOI Detection.** To overcome the limited granularity of CLIP features, several works explore LLMs and MLLMs. CMD-SE (Lei et al., 2024b) and UniHOI (Cao et al., 2023) generate textual descriptions of interactions, while BC-HOI (Hu et al., 2025) employs MLLM-based captioning loss to supervise training. Other efforts, such as RLIP (Yuan et al., 2022; 2023) and MP-HOI (Rombach et al., 2022), expand training data via relational pretraining or generative augmentation. Although these method exploit auxiliary signals from LLMs or MLLMs, they still rely heavily on CLIP features and remain entangled with specific detectors.

We depart from prior work by fully decoupling interaction recognition from object detection. Instead of relying on CLIP embeddings, we directly leverage MLLMs for interaction recognition. By reformulating the task as VQA with proposed deterministic generation, our framework enables training-free zero-shot prediction. Furthermore, our spatial-aware pooling and one-pass deterministic matching methods explicitly address robustness and efficiency issues through model fine-tuning, setting our method apart from both CLIP-based and MLLM-based prior works.

## 3 METHODOLOGY

### 3.1 OVERVIEW

Given an image $I$, HOI detection aims to locate humans and objects, and recognizes the interactions (verbs) between each human-object pair. Conceptually, the interaction can be defined as a triplet $\{B_h, B_o, (A, C_o)\}$, where $B_h$ and $B_o$ denote the bounding box of the human and object, respectively, $C_o \in \mathbb{C} = \{c_1, c_2, \ldots, c_{N_o}\}$ denotes the object category, and $A \in \mathbb{A} = \{a_1, a_2, \ldots, a_{N_a}\}$ denotes the interaction category. Traditional HOI detection assumes the $\mathbb{C}$, $\mathbb{A}$, and their combination appear in the training set. Differently, in zero-shot HOI detection, some combinations of object and verb (Unseen Composition, UC), some categories of objects (Unseen Object, UO), and some categories of verbs (Unseen Verb, UV) would disappear.

The progress in open-vocabulary object detection (Liu et al., 2024; Cheng et al., 2024) provides a promising solution for object localization in HOI detection. In contrast, interaction recognition (IR), especially for unseen interactions, remains a challenging task. To alleviate the above restrictions, we fully decouple the object detection and IR, and leverage the powerful Multi-modal Large Language Models (Liu et al., 2023; Bai et al., 2025) (MLLMs) for zero-shot IR. Specifically, given detection results of any detector $\{C^i, B^i\}_{i=1}^{N_{\text{det}}}$, where $C^i$ denotes the object category, we associate every human

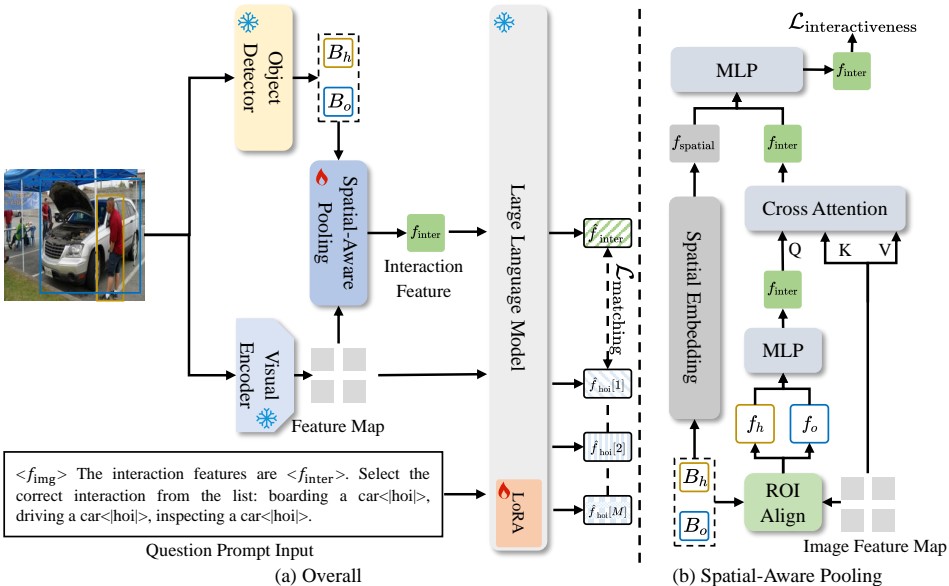

Figure 2: The overall framework of our method and the spatial-aware pooling (SAP). (a) The proposed method decouples the object detection and interaction recognition for HOI detection. With the detected human-object pair, a MLLM is used to recognize their interaction. To enhance both performance and inference efficiency, SAP integrates appearance and pairwise spatial cues, and a one-pass deterministic matching method enables the prediction of all candidate interactions in a single forward pass. (b) SAP takes the human and object features as input. The cross attention layer aggregates features beyond the area of bounding box, enhancing robust to the noise in the detection results. Spatial Embedding encodes the useful pairwise information into the interaction features.

instance with each object instance to formulate the human-object pairs,

$$\{B_h^i, B_o^i, C_o^i\}_{i=1}^{N_{\text{inter}}} \leftarrow \{(B^j, C^j, B^k, C^k) \,|\, j \neq k, C^j = \text{human}\}_{j=[1:N_{\text{det}}], k=[1:N_{\text{det}}]}, \qquad (1)$$

where $C_o^i$ is the object category of the $i$-th human-object pair. For each human-object pair, a MLLM is used to predict the confidence score $\{S_v^i\}_{i=1}^{N_{\text{inter}}} = \{\text{MLLM}(B_h^i, B_o^i, C_o^i, I)\}_{i=1}^{N_{\text{inter}}}$, where $S_v^i \in \mathbb{R}^M$ indicates the probability of the human-object pair having corresponding interaction, $M$ is the number of interaction categories. In the following parts, we will first introduce the MLLM-based training-free IR, and further propose a tuning method to break through the bottlenecks of MLLMs for IR. The overall framework is shown in Fig. 2.

## 3.2 MLLM-BASED TRAINING-FREE INTERACTION RECOGNITION

MLLMs are constructed with a visual encoder $f_{\text{img}} = \Phi_V(I)$, which extracts the features of a given image, and an LLM. MLLMs employ visual question answering (VQA) to address various tasks: Given a question prompt $Q$ and $f_{\text{img}}$, they are formulated as the prefix input for the LLM to generate the answer. The final result can be obtained through parsing the answer. To cast IR as a VQA task, given an associated human-object pair $(B_h^i, B_o^i, C_o^i)$, we construct a question prompt $Q^i$ that asks the MLLM to recognize the interaction:

*Question*: $<f_{\text{img}}>$ The interaction features are $<f_{\text{inter}}^i>$. Select the correct interaction from the list: $<\Theta(C_o^i)>$.

The $<f_{\text{img}}>$ is replaced with $f_{\text{img}}$. To guide the MLLM focus on a specific human-object pair, we extract human and object features through ROIAlign (He et al., 2017) and concatenate them to construct interaction features:

$$\begin{aligned} f_h^i, f_o^i &= \text{ROIAlign}(f_{\text{img}}, [B_h^i, B_o^i]), \\ f_{\text{inter}}^i &= \text{Concat}(f_h^i, f_o^i). \end{aligned} \qquad (2)$$

To achieve open-vocabulary IR, we define the candidate interaction list $\Theta(C_o^i)$ based on the category of detected object, *e.g.*, *feeding a bird, chasing a bird, holding a bird*. This list gives a hint to prompt the MLLM to select the correct interaction, *e.g., feeding a bird, holding a bird*.

**Deterministic Generation.** Although MLLM shows strong generalization ability for various tasks, directly using the above pipeline leads to poor performance. The reasons are: a) IR is a multi-label classification task, *i.e.*, a human-object pair has multiple interactions. b) The metric mAP requires the model to give a confidence score to the prediction. The model can give answers with various format and tends to generate one interaction even after supervised fine-tuning as shown in Table 3. Moreover, the confidence score is hard to obtain.

To overcome these limitations, we change the open-ended text generation of the MLLM to deterministic generation proxy task. It leverages the discriminative capability of MLLMs by estimating the semantic similarity between the question prompt and the candidate interaction. Specifically, given the candidate interaction list of the detected object in a text format: $\Theta(C_o^i) = \{T_1, T_2, \ldots, T_M\}$, we measure the semantic similarity between the prompt question and the interaction by calculating the conditional likelihood for the MLLM to generate the corresponding interaction,

$$S_v^i[k] = p(T_k|I, Q^i) = \prod_{j=1}^{N} p(t[j]|T_k[<j], I, Q^i), k = 1, 2, \ldots, M, \tag{3}$$

where $Q^i$ is the question prompt, $t[j]$ is the $j$-th token in the answer $T_k$, and $S_v^i[k]$ is the semantic similarity between $Q^i$ and the interaction $T_k$, which can be used as the prediction confidence score. Deterministic generation prevents the MLLM from generating answers unrelated to the interaction. As shown in the experiments, this method significantly improves the performance in a training-free manner. To further improve the performance, we can perform supervised fine-tuning on the dataset. Note that deterministic generation is still essential.

## 3.3 SPATIAL-AWARE POOLING

The proposed training-free IR method exploits the strong generalization ability of MLLMs and achieves competitive performance. However, there are still some bottlenecks: a) The interaction features are obtained with ROIAlign based on the detected bounding box. The pooling features are restricted within the bounding box, making it sensitive to the noise in the detection results, *e.g.*, the bounding box may cover a part of the object or include some background. Moreover, it ignores the human-object pairwise spatial information that is proved to be important for IR (Zhang et al., 2021; 2022a). b) The computational complexity is high. Calculating the confidence score of an interaction requires a forward pass as in Eq. 3. Given an interaction list $\{T_1, T_2, \ldots, T_M\}$ with $M$ candidate interactions, $M$ forward passes are needed.

To mitigate the impact of imperfect detection results and integrate pairwise spatial information, we propose spatial-aware pooling (SAP). For simplicity, we omit the upperscript of the human-object pair. Specifically, as shown in Fig. 2 (b), given bounding boxes $B_h, B_o$ of a human-object pair, SAP takes the pooled human and object features $f_h, f_o$ as the input. Then these features are merged through a Multilayer Perceptron (MLP) to formulate the interaction features $f_{\text{inter}}$. A cross attention layer aggregates useful information from the image features into $f_{\text{inter}}$. Similar to Zhang et al. (2022a), assume $B_h = [x_h, y_h, w_h, h_h]$, where $x_h$ and $y_h$ denote the center of the bounding box while $w_h$ and $h_h$ are the width and height, we leverage the box areas, aspect ratios, intersection over union (IoU), and direction from human to the object to form the pairwise spatial vector,

$$U = [w_h h_h, w_o h_o, \frac{w_h}{h_h}, \frac{w_o}{h_o}, \text{IoU}(B_h, B_o), \frac{x_h - x_o}{w_h}, \frac{y_h - y_o}{h_h}],$$
$$f_{\text{spatial}} = \Phi_{\text{spatial}}(U), \tag{4}$$

where $\Phi_{\text{spatial}}(U)$ is a MLP to project the spatial vector into the space of $f_{\text{inter}}$. The final MLP merges spatial information in the spatial features into the interaction features.

Cross attention operation between the interaction features and image features allows the information aggregation beyond the area of bounding box, relieving the impact of imperfect detection results as shown in Fig. 5. The output $f_{\text{inter}}$ of spatial-aware pooling contains both appearance and spatial information of a given human-object pair, making it suitable for IR. The new interaction features are used to replace $<f_{\text{inter}}>$ in the question prompt through a projection layer.

Our method associates every human instance with every object instance, which leads to a large number of non-interactive human-object pairs. Filtering out non-interactive human-object pairs before sending them into the LLM can significantly reduce the computational complexity. Therefore, based on the interaction features, we add a one linear classifier to predict the interactiveness of a human-object pair,

$$S_{\text{interactiveness}} = \sigma(\text{Linear}(f_{\text{inter}})), \tag{5}$$

where $S_{\text{interactiveness}}$ is a $[0, 1]$ scalar that measures whether a person and an object are interacting, $\sigma$ is the sigmoid function.

### 3.4 ONE-PASS DETERMINISTIC MATCHING

With Eq. 5, most of non-interactive human-object pairs can be filtered out. However, for each human-object pair, multiple forward passes of the LLM are still needed to calculate the interaction scores for all candidate interactions with Eq. 3. To further reduce the computational complexity, we propose a one-pass deterministic matching method to change the deterministic generation as a feature matching task. Specifically, for a human-object pair $(B_h^i, B_o^i, C_o^i)$, and its corresponding candidate interaction list $\Theta(C_o^i) = \{T_1, T_2, \ldots, T_M\}$, we add a special token $< |\text{hoi}| >$ after each candidate interaction, and change the question prompt to:

---

*Question*: $<f_{\text{img}}>$ The interaction features are $<f_{\text{inter}}^i>$. Select the correct interaction from the list: $T_1< |\text{hoi}| >, T_2< |\text{hoi}| >, \ldots, T_M< |\text{hoi}| >$.

---

After tokenizing the question prompt $Q$ and replacing the placeholders $<f_{\text{img}}>$, $<f_{\text{inter}}^i>$ with corresponding image features and interaction features, respectively, we send the prompt features into the LLM for feature extraction. Assuming the output features of each special token are $\hat{f}_{\text{hoi}}^i[k]$, where $k$ is the index of the candidate interaction in the list, the output interaction features are $\hat{f}_{\text{inter}}^i$, the cosine similarity between these two features is used as the confidence score,

$$S_{\text{v}}^i[k] = \text{cosine}(\hat{f}_{\text{hoi}}^i[k], \hat{f}_{\text{inter}}^i), k = 1, 2, \ldots, M. \tag{6}$$

With Eq. 6, for each human-object pair, we can predict all candidate interactions with only one forward pass of the LLM through features matching.

### 3.5 TRAINING AND INFERENCE

**Training.** We train our method with two stages. At the first stage, we train SAP only. Specifically, given a set of associated human-object pairs, we assign a positive label for a human-object pair if the bounding boxes of both human and object have an IoU exceeding a threshold with the ground truth. The binary focal loss (Lin et al., 2017) is adopted,

$$\mathcal{L}_{\text{interactiveness}} = \frac{1}{N_{\text{inter}}} \sum_{i=1}^{N_{\text{inter}}} \text{FocalBCE}(S_{\text{interactiveness}}^i, Y_{\text{interactiveness}}^i), \tag{7}$$

where $Y_{\text{interactiveness}}^i \in \{0, 1\}$ is the interactiveness label.

At the second stage, we freeze SAP, and train the LLM with LoRA Hu et al. (2022). Specifically, by assigning the human-object pair with a ground truth interaction pair, the labels are determined by the order of the positive interaction in the candidate interaction list of the question prompt. For example, if the interactions of the ground truth interaction pair are *feeding a bird, holding a bird*, the candidate interaction list in the question prompt is *feeding a bird, chasing a bird, holding a bird*, the interaction labels are $Y_{\text{v}}^i = [1, 0, 1]$. The binary focal loss is adopted,

$$\mathcal{L}_{\text{matching}} = \frac{1}{N_{\text{inter}}|\Theta(C_o^i)|} \sum_{i=1}^{N_{\text{inter}}} \sum_{k=1}^{|\Theta(C_o^i)|} \text{FocalBCE}(S_{\text{v}}^i[k], Y_{\text{v}}^i[k]), \tag{8}$$

where $k$ is the $k$-th interaction in the candidate interaction list $\Theta(C_o^i)$, $|\Theta(C_o^i)|$ denotes the number of interaction in the list. $S_{\text{v}}^i[k]$ is obtained with Eq. 6 and clamped into $[0, 1]$ to avoid negative value. Note that, the visual encoder in the MLLM is frozen in both training stages.

Table 1: Performance comparison on HICO-DET under zero-shot, cross-detector and training-free settings. ⋄ indicates BLIP2 (Li et al., 2023) features is used. † and ‡ indicate Grounding-DINO (Liu et al., 2024) and Yolo-World (Cheng et al., 2024) are used as the detector, respectively. ⋆ is the training-free setting.

| Methods | RF-UC | | | NF-UC | | | UO | | | UV | | | Avg |
| | Unseen | Seen | Full | Unseen | Seen | Full | Unseen | Seen | Full | Unseen | Seen | Full | Full |
|---|---|---|---|---|---|---|---|---|---|---|---|---|---|
| ADA-CM | 27.63 | 34.35 | 33.01 | 32.41 | 31.13 | 31.39 | - | - | - | - | - | - | - |
| ADA-CM† | 20.82 | 30.12 | 28.26 | 26.23 | 27.03 | 26.87 | - | - | - | - | - | - | - |
| ADA-CM‡ | 20.43 | 29.23 | 27.47 | 24.89 | 27.69 | 27.13 | - | - | - | - | - | - | - |
| BCOM | 28.52 | 35.04 | 33.74 | 33.12 | 31.76 | 32.03 | - | - | - | - | - | - | - |
| BCOM† | 9.23 | 23.08 | 20.31 | 16.38 | 20.73 | 19.86 | - | - | - | - | - | - | - |
| BCOM‡ | 8.17 | 20.07 | 17.69 | 13.76 | 17.81 | 17.00 | - | - | - | - | - | - | - |
| GEN-VLKT | 21.36 | 32.91 | 30.56 | 25.05 | 23.38 | 23.71 | 10.51 | 28.92 | 25.63 | 20.96 | 30.23 | 28.74 | 27.16 |
| HOICLIP | 25.53 | 34.85 | 32.99 | 26.39 | 28.10 | 27.75 | 16.20 | 30.99 | 28.53 | 24.30 | 32.19 | 31.09 | 30.09 |
| CLIP4HOI | 28.47 | 35.48 | 34.08 | 31.44 | 28.26 | 28.90 | 31.79 | 32.73 | 32.58 | 26.02 | 31.14 | 30.42 | 31.50 |
| CMMP | 29.45 | 32.87 | 32.18 | 32.09 | 29.71 | 30.18 | 33.76 | 31.15 | 31.59 | 26.23 | 32.75 | 31.84 | 31.45 |
| LAIN | 31.83 | 35.06 | 34.41 | 36.41 | 32.44 | 33.23 | 37.88 | 33.55 | 34.27 | 28.96 | 33.80 | 33.12 | 33.76 |
| EZ-HOI | 34.24 | 37.35 | 36.73 | 36.33 | 34.47 | 34.84 | 38.17 | 36.02 | 36.38 | 28.82 | 38.15 | 36.84 | 36.20 |
| UniHOI⋄ | 28.68 | 33.16 | 32.27 | 28.45 | 32.63 | 31.79 | 19.72 | 34.76 | 31.56 | 26.05 | 36.78 | 34.68 | 32.08 |
| BC-HOI⋄ | 42.31 | 40.67 | 40.99 | 33.01 | 37.24 | 36.40 | 19.94 | 37.03 | 34.18 | 31.18 | 41.31 | 39.89 | 37.87 |
| Ours | 41.79 | 44.01 | 43.56 | 43.12 | 39.63 | 40.33 | 48.67 | 42.58 | 43.60 | **36.89** | 43.84 | 42.88 | 42.59 |
| Ours† | 43.30 | **45.19** | **44.81** | 41.52 | 41.51 | 41.51 | **50.15** | 44.31 | 45.28 | 36.88 | **45.66** | 44.43 | **44.00** |
| Ours‡ | **43.81** | 44.05 | 44.00 | **43.36** | **41.67** | 42.01 | 47.95 | 44.19 | 44.82 | 36.25 | 45.13 | 43.88 | 43.68 |
| ADA-CM⋆ | - | - | 25.19 | - | - | 25.19 | - | - | 25.19 | - | - | 25.19 | 25.19 |
| Ours⋆ | - | - | **31.50** | - | - | **31.50** | - | - | **31.50** | - | - | **31.50** | **31.50** |

**Inference.** During inference, with the detection results of any detector, we first formulate the human-object pairs, and leverage SAP to filter out the non-interactive pairs whose $S_{\text{interactiveness}}$ is below a threshold $\lambda$, the remaining human-object pairs are sequentially sent into the MLLM for IR. The final interaction confidence score of the $i$-th human-object pair is,

$$\hat{S}_v^i[k] = S_v^i[k] \cdot S_{\text{interactiveness}}^i \cdot S_h^i \cdot S_o^i, k = 1, 2, \ldots, M \tag{9}$$

where $S_h^i$ and $S_o^i$ are the detection score given by the detector. Our method fully decouples the object detection and IR, and can combine with any detector without retraining.

## 4 EXPERIMENTS

### 4.1 EXPERIMENTAL SETTINGS

**Datasets.** We evaluate our method on two widely used HOI detection benchmarks: HICO-DET (Chao et al., 2018) and V-COCO (Gupta & Malik, 2015). HICO-DET (Chao et al., 2018) contains 47,776 images annotated with 600 HOI categories, defined over 80 object categories and 117 verbs. The default setting splits the dataset with 38,118 images for training and 9,658 for testing. V-COCO (Gupta & Malik, 2015) is built upon MS COCO and provides annotations for 29 action categories across 10,396 images. We adopt the official train/val/test splits.

**Data Structure and Evaluation Metrics for Zero-Shot, Cross-Detector and Cross-Dataset Settings.** To evaluate generalization to unseen HOI categories, we follow the settings introduced in prior works (Bansal et al., 2020; Hou et al., 2020) on HICO-DET: Rare First Unseen Combination (RF-UC): unseen HOIs are constructed by holding out rare verb–object pairs during training. Non Rare First Unseen Combination (NF-UC): unseen HOIs are constructed by holding out non-rare verb–object pairs. Unseen Verb (UV): unseen HOIs involve verbs never observed during training. Unseen Object (UO): unseen HOIs involve objects never observed during training. To validate the decouple framework, the cross-detector setting uses the detection results with various detectors without retraining the model. In addition, we consider a cross-dataset setting, training on HICO-DET and testing on V-COCO, which poses a more challenging generalization scenario. We report mean Average Precision (mAP) as an evaluation metric. An HOI triplet is considered correct if the human and object boxes achieve IoU $\geq 0.5$ with ground truth and the interaction category is correct.

**Implementation Details.** For the object detector, we adopt ResNet50 DETR (Carion et al., 2020), Grounding-DINO (Liu et al., 2024), and YoLo-World (Cheng et al., 2024). The MLLM is Qwen 2.5-VL 3B (Bai et al., 2025). Training-free inference uses our proposed deterministic generation method. Fine-tuning is conducted with AdamW (Loshchilov & Hutter, 2017). At the first stage, we train SAP for 30 epochs with a learning rate of 0.0001 and a batch size of 16. At the second stage,

we only fine-tune the LoRA in the LLM for 16 epochs with a learning rate of 0.0001 and a batch size of 16. We use a similar data augmentation with most of HOI detection methods (Zhang et al., 2022a; 2023; Ning et al., 2023). During training, we use the ground truth bounding box of objects as the detection results. During inference, the human-object pairs whose interaction score below 0.15 is filtered out. All experiments are conducted on 4 Nvidia RTX 3090s.

## 4.2 COMPARISON WITH OTHER METHODS

**Zero-shot Settings.** Table 1 summarizes the comparison with other methods on HICO-DET under four zero-shot settings. The comparison methods include one-stage methods: GEN-VLKT (Liao et al., 2022), HOICLIP (Ning et al., 2023), two-stage methods: ADA-CM (Lei et al., 2023), CLIP4HOI (Mao et al., 2024), BCOM (Wang et al., 2024), CMMP (Lei et al., 2025), LAIN (Kim et al., 2025), and methods that leverage BLIP2 (Li et al., 2023) or MLLM: UniHOI (Cao et al., 2023), BC-HOI (Hu et al., 2025), EZ-HOI (Lei et al., 2024a). Our method consistently achieves the best performance across all settings. Its generalization ability on unseen interactions is impressive, *e.g.*, it surpasses BC-HOI by about 10.11% and 28.73% on the NF-UC and UO settings.

**Cross-Detector Setting.** Once trained, our framework allows the detector to be freely changed without retraining on HICO-DET. When combined with Grounding-DINO or Yolo-World, our method further improves the average mAP to 44.00% and 43.68%. This detector-agnostic property highlights the modularity of our method: stronger object detectors can directly boost HOI detection performance. In contrast, most methods rely on the features from a specific detector. Only ADA-CM and BCOM do not rely on detector features. However, they still cannot change the detectors without retraining, *e.g.*, replacing the original ResNet-50 DETR with other detectors (Grounding-DINO or Yolo-World) leads to degraded performance. BCOM leverages UPT that models the relationship between different detected objects, while ADA-CM introduces an instance-aware adapter that uses the prior knowledge of detected objects. With these designs, the model training relies on the detection results of a specific detector, and inter-object relations, leading to the coupling of the detector and the interaction recognizer.

**Training-Free Setting.** Our method achieves 31.50% mAP, outperforming ADA-CM by a clear margin. Even without any training, our method achieves performance comparable to methods that rely on complex training strategy, *e.g.*, UniHOI, CMMP.

**Cross-Dataset Setting.** To further validate the generalization of our method, we conduct a more challenging cross-dataset setting, where the model is trained on HICO-DET and evaluated on V-COCO. As shown in Table 2, our method achieves a substantial improvement over prior

Table 2: Performance comparison on V-COCO under cross-dataset setting.

| Methods | $H \rightarrow$ V-COCO $mAP_{role}^{\#2}$ |
|---|---|
| GEN-VLKT | 42.34 |
| ADA-CM | 46.39 |
| HOICLIP | 41.82 |
| BCOM | 48.87 |
| CMMP | 47.65 |
| LAIN | 48.34 |
| Ours | **59.91** |

works, reaching 59.91% $mAP_{role}^{\#2}$. These promising results demonstrate the strong generalization of leveraging MLLMs for IR, while the detector-agnostic design makes our method both highly flexible and broadly applicable.

## 4.3 ABLATION STUDY

**Effect of Deterministic Generation.** To evaluate the effect of deterministic generation, we conduct experiments under the training-free setting with different formulations of question prompts in Table 3. "Simple" is the simple QA format. "Multiple Choice Questions" denotes formulating the candidate interactions as multiple choices and the model should give the correct choices. "In Context"denotes some example QA pairs are given in the prompt. The open-ended text generation procedure of the MLLM leads to a high format error rate. More importantly, the model has a strong bias towards single-interaction predictions, leading to low interaction recognition performance. Leveraging conditional likelihood as a confidence score brings limited improvement. In contrast, our proposed deterministic generation eliminates format errors and ensures multi-interaction outputs, boosting the mAP to 31.50%. Moreover, deterministic generation is also important to the model after supervised fine-tuning (SFT), improving the performance from 31.61% to 39.87%. These results demonstrate that deterministic generation is essential for both training-free and trained models.

Table 3: Ablation studies on the deterministic generation. "Simple", "Multiple Choice Questions", "In Context" are the different formulations of question prompts. Some examples are shown in A.5. "*" denotes leveraging the conditional likelihood of the answer as the confidence score of the prediction. "Format Error Rate" denotes the rate at which the format of the answer is error. "Single Output Rate" indicates the rate at which the answer only contains one interaction. "SFT" indicates the model fine-tuned on the HICO-DET with supervised fine-tuning.

| Settings | Full | Format Error Rate | Single Output Rate |
|---|---|---|---|
| Simple | 14.23 | 36.78 | 80.91 |
| Multiple Choice Questions | 18.91 | 23.75 | 80.17 |
| In Context | 19.57 | 18.75 | 78.82 |
| Simple* | 16.19 | 36.78 | 80.91 |
| Multiple Choice Questions* | 20.36 | 23.75 | 80.17 |
| In Context* | 20.97 | 18.75 | 78.82 |
| Deterministic Generation | 31.50 | 0 | 0 |
| SFT | 31.61 | 6.27 | 58.23 |
| SFT + Deterministic Generation | 39.87 | 0 | 0 |

Table 4: Ablation studies on each proposed component. The model trained with SFT and inference with deterministic generation serves as the baseline. "SAP" and "DM" are the spatial-aware pooling module and deterministic matching, respectively. "Pairwise Spatial" and "Cross Attention" are the integration of pairwise spatial information and cross attention to aggregate image features, respectively. "UPT" indicates the unary-pairwise transformer is used instead of SAP.

| Methods | UO | | | UV | | | Inference |
| | Unseen | Seen | Full | Unseen | Seen | Full | Time ($ms$) |
|---|---|---|---|---|---|---|---|
| Baseline | 43.66 | 38.36 | 39.24 | 34.06 | 38.46 | 37.84 | 569 |
| w/ SAP | 45.91 | 41.59 | 42.31 | 36.47 | 42.84 | 41.95 | 217 |
| w/ DM | 45.63 | 39.47 | 40.50 | 36.14 | 39.75 | 39.24 | 189 |
| w/o Pairwise Spatial | 44.31 | 41.52 | 41.98 | 35.32 | 41.66 | 40.77 | 86 |
| w/o Cross Attention | 43.61 | 40.92 | 41.37 | 34.50 | 41.76 | 40.74 | 87 |
| UPT | 43.66 | 41.38 | 41.76 | 35.29 | 41.44 | 40.58 | 122 |
| Ours | 48.67 | 42.58 | 43.60 | 36.89 | 43.84 | 42.88 | 91 |

**Effect of Each Proposed Component.** Table 4 analyzes the contributions of spatial-aware pooling (SAP) and deterministic matching (DM). The inference time denotes the average inference time per image. With SAP, most of non-interaction human-object pairs are filtered out. DM further enables the prediction of multiple candidate interactions in a single forward pass. These two components bring clear improvements over the baseline and reduce the computational complexity.

We also evaluate the design of SAP. Removing either pairwise spatial encoding or cross attention results in noticeable drops in both UO and UV settings. Specifically, discarding spatial encoding reduces the ability to reason about relative human-object positions, while discarding cross attention weakens the robustness to noise in the detection results. The unary-pairwise transformer (Zhang et al., 2022a) is widely used by many methods (Lei et al., 2025; Mao et al., 2024) to integrate the pairwise spatial and appearance information. We replace SAP with UPT, and change its input with the human and object features obtained by ROIAlign. This setting leads to a performance drop and a coupling of the detector as UPT leverages the relationship between different detected objects.

**Impact of Candidate Order.** To investigate the effect of candidate ordering in the question prompt, we perform inference 5 times using different permutations of the candidate list. As shown in Table 5, our method remains robust to different candidate orders, with only minor performance fluctuations.

Table 5: Impact of the Candidate Order.

| Unseen | Seen | Full |
|---|---|---|
| 48.65 ± 0.03 | 42.59 ± 0.02 | 43.61 ± 0.02 |

## 5 CONCLUSION

This paper proposes a decoupled framework for zero-shot HOI detection, which separates object detection from interaction recognition (IR), and exploits the powerful multi-modal large language models (MLLMs) for IR. By casting IR as a VQA task, our method achieves superior performance in the training-free setting with deterministic generation. To further overcome the bottlenecks of MLLMs for IR, we introduce a spatial-aware pooling module that integrates pairwise spatial information, and a one-pass deterministic matching method that reduces the computational complexity by converting the generation to a feature matching task. Extensive experiments on HICO-DET and V-COCO demonstrate the effectiveness of our method, which achieves impressive performance under zero-shot and cross-dataset settings. Once trained, our method has the flexibility to integrate with any object detector without retraining and can benefit from the progress of detectors.

## ACKNOWLEDGEMENT

This work is supported in part by Natural Science Foundation of China under Grant No. 62425603, 62506167, 62502397, in part by the Basic Research Program of Jiangsu Province under Grant No. BK20240011, BK20251451, in part by the Fundamental Research Funds for the Central Universities under Grant No. 30925010206, in part by Natural Science Foundation of Sichuan Province under Grant No. 2026NSFSC1449.

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
