# A APPENDIX

## A.1 USAGE OF LLMS

We employed a large language model (LLM) as a writing assistant to improve the clarity, readability, and presentation of our paper. Specifically, the LLM was used to polish grammar, refine phrasing, and reorganize text for better readability. All core research ideas, technical designs, experimental implementations, analyses, and conclusions were conceived and developed entirely by the authors. The authors take full responsibility for the content of the paper.

## A.2 ETHICS STATEMENT

This work focuses on zero-shot human–object interaction detection by leveraging MLLMs. Our experiments are conducted on publicly available datasets (HICO-DET, and V-COCO), which are widely used benchmarks in the research community. We strictly follow their licensing terms and do not collect or annotate new data. HOI detection involves the analysis of human activities in visual data, which could be misused in sensitive domains. We encourage responsible use of this research and emphasize that applications should comply with legal and ethical standards, particularly regarding privacy, fairness, and non-discrimination.

Table 6: Ablation studies on training settings. "Full Tuning" and "LoRA" indicate full tuning or LoRA tuning of the model, respectively. "Visual" and "LLM" denote tuning the visual encoder or LLM of MLLM, respectively.

| Methods | UO | | | UV | | |
|---|---|---|---|---|---|---|
| | Unseen | Seen | Full | Unseen | Seen | Full |
| Full Tuning LLM | 47.59 | 43.03 | 43.79 | 35.39 | 43.67 | 42.51 |
| Full Tuning Visual + LLM | 46.47 | 42.78 | 43.39 | 34.10 | 43.05 | 41.80 |
| LoRA Visual + LLM | 48.26 | 42.74 | 43.66 | 35.66 | 43.38 | 42.30 |
| Ours | 48.67 | 42.58 | 43.60 | 36.89 | 43.84 | 42.88 |

## A.3 MORE EXPERIMENTAL RESULTS

**Effect of Different Training Settings.** As shown in Table 6, comparing different training settings, fully tuning the LLM or the entire model provides moderate gains, but comes at higher training cost. In contrast, lightweight LoRA tuning achieves comparable or even better performance to full tuning.

**Effectiveness on Different MLLMs.** We further test our method with different MLLMs, including LLaVA-OneVision 0.5B (Li et al., 2024a), Qwen2.5-VL 3B, and Qwen2.5-VL 7B. As shown in Table 7, our method consistently improves as the capacity of the underlying MLLM increases. While the lightweight LLaVA-OneVision 0.5B already achieves competitive results, larger-scale models such as Qwen2.5-VL 7B bring notable gains across both UO and UV settings, demonstrating that our method can be directly applied to different MLLMs.

**Performance on Fully Supervised Setting.** To further verify the effectiveness of our framework on fully supervised setting, we conduct experiments on both HICO-DET and V-COCO. Results are summarized in Table 8. Compared to existing method, our method achieves clear improvements across all metrics, reaching 44.58% mAP on HICO-DET and 71.7% $mAP_{role}^{\#2}$ on V-COCO. On Fully supervised setting, the performance can also be improved when combined with advanced open-

Table 7: Ablation studies on different MLLMs.

| Methods | UO | | | UV | | |
|---|---|---|---|---|---|---|
| | Unseen | Seen | Full | Unseen | Seen | Full |
| LLaVA Onevision 0.5B | 45.29 | 41.34 | 42.00 | 34.35 | 42.29 | 41.18 |
| Qwen 2.5-VL 3B | 48.67 | 42.58 | 43.60 | 36.89 | 43.84 | 42.88 |
| Qwen 2.5-VL 7B | 50.78 | 45.03 | 45.99 | 38.73 | 45.77 | 44.78 |

Table 8: Performance comparison on HICO-DET and V-COCO under fully supervised setting. ◇ indicates BLIP2 (Li et al., 2023) features is used. † and ‡ indicate Grounding-DINO (Liu et al., 2024) and Yolo-World (Cheng et al., 2024) are used as the detector, respectively.

| Methods | HICO-DET | | | V-COCO | |
|---|---|---|---|---|---|
| | Full | Rare | Non-Rare | $\text{mAP}_{role}^{\#1}$ | $\text{mAP}_{role}^{\#2}$ |
| GEN-VLKT | 33.75 | 29.25 | 35.10 | 62.4 | 64.5 |
| HOICLIP | 34.69 | 31.12 | 35.74 | 63.5 | 64.8 |
| CLIP4HOI | 35.33 | 33.95 | 35.74 | - | 66.3 |
| ADA-CM | 33.80 | 31.72 | 34.42 | - | 61.2 |
| CMMP | 38.14 | 37.75 | 38.25 | - | 64.0 |
| BCOM | 39.34 | 39.90 | 39.17 | 65.8 | 69.9 |
| LAIN | 36.02 | 35.70 | 36.11 | - | 65.1 |
| EZ-HOI | 38.61 | 37.70 | 38.89 | 60.5 | 66.2 |
| UniHOI◇ | 40.06 | 39.91 | 40.11 | 65.6 | 68.3 |
| BC-HOI◇ | 43.01 | 45.76 | 42.18 | 68.2 | 70.6 |
| Ours | 44.58 | 46.17 | 44.08 | 69.4 | 71.7 |
| Ours† | **46.21** | 47.54 | **45.81** | **70.4** | **72.5** |
| Ours‡ | 45.62 | **49.34** | 44.51 | 70.1 | 72.4 |

Table 9: Performance comparison with ground-truth detection results. ADA-CM and BCOM are trained with ResNet50 DETR as the detector. "Ground-Truth" indicates the annotated bounding box is used as the detection results.

| Settings | RF-UC | | | NF-UC | | |
|---|---|---|---|---|---|---|
| | Unseen | Seen | Full | Unseen | Seen | Full |
| ADA-CM | 27.63 | 34.35 | 33.01 | 32.41 | 31.13 | 31.39 |
| w/ Ground-Truth | 26.89 | 35.50 | 33.78 | 30.13 | 31.18 | 30.97 |
| BCOM | 28.52 | 35.04 | 33.74 | 33.12 | 31.76 | 32.03 |
| w/ Ground-Truth | 10.78 | 20.83 | 18.82 | 17.60 | 20.16 | 19.65 |
| Ours | 41.79 | 44.01 | 43.56 | 43.12 | 39.63 | 40.33 |
| w/ Ground-Truth | 65.22 | 65.14 | 65.16 | 58.65 | 62.30 | 61.57 |

vocabulary detectors. The above results further demonstrate the strong flexibility and robustness of our method.

**Performance with Ground-Truth Detection Results.** Our framework fully decouples object detection from interaction recognition, enabling recognition of any interaction given any human–object pair. In many practical scenarios, users may only be concerned with the interaction of specific human–object pairs. To this end, we evaluate our framework using annotated bounding boxes as detection results and compare it with ADA-CM and BCOM under the same setting. Results are shown in Table 9. Both ADA-CM and BCOM experience severe performance degradation with ground-truth detections, indicating that their interaction recognition modules are heavily entangled with a specific detector and fail to generalize once the detection pipeline is altered. In contrast, our method demonstrates a substantial performance gain, improving from 43.56% to 65.16% mAP on RF-UC and from 40.33% to 61.57% mAP on NF-UC. This ability further broadcasts our method across different evaluation settings.

**Inference Time.** We provide a detailed inference-time comparison with other methods. All experiments are conducted on a single NVIDIA RTX 3090 GPU with a batch size of 1. For two-stage methods, we also report the detection and IR inference times. As shown in Table 10, our method achieves competitive inference speed while delivering significantly stronger performance. Importantly, our decoupled framework enables plug-and-play detector replacement, allowing users to flexibly trade off speed and accuracy by choosing an appropriate detector.

## A.4 QUALITATIVE RESULTS

**Visualization of Successful Cases.** We present several successful HOI detection results in Fig. 3. In the 1st and 4th examples, multiple humans interact with the same object, and our method accurately

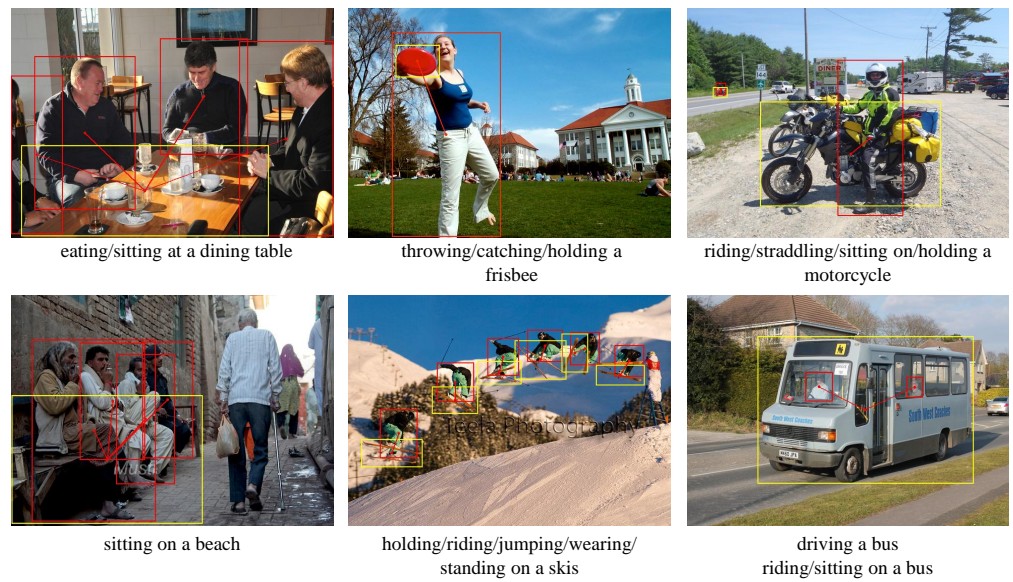

Figure 3: Visualization of successful examples. Humans (subject) are marked with red rectangles, and objects with yellow rectangles.

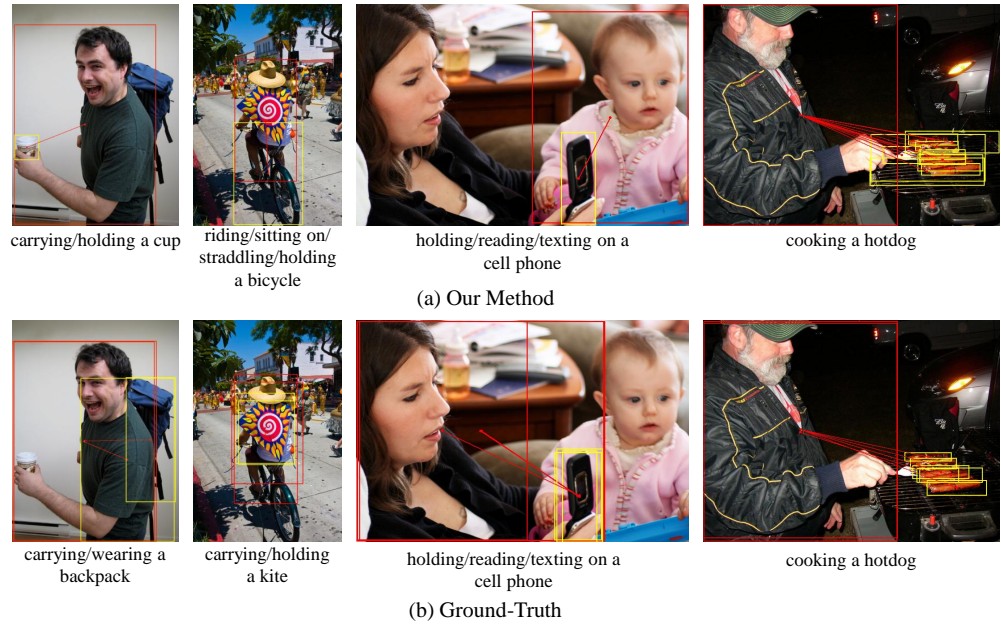

Figure 4: Visualization of failure cases. Humans (subject) are marked with red rectangles, and objects with yellow rectangles.

identifies all interactive pairs and their corresponding interactions. In the 3rd and 6th examples, our method shows ability to identify tiny human-object pairs and recognize their interaction. These visualizations highlight the robustness of our framework across challenging scenarios.

**Visualization of Failure Cases.** To better understand the limitations of our method, we show typical failure cases in Fig. 4. In the 1st and 2nd examples, our predictions are correct, but the ground-truth annotations miss some valid human–object pairs, leading to false positives. In the 3rd example, our method incorrectly associates the baby with the cell phone instead of the woman, who is the true

Table 10: Comparisons in inference time.

| Methods | Inference Time (*ms*) | *Avg* mAP |
|---|---|---|
| UniHOI | 86 | 32.08 |
| BC-HOI | 102 | 37.87 |
| ADA-CM | 93 (27 + 66) | - |
| EZ-HOI | 100 (27 + 73) | 36.20 |
| Ours + ResNet50 DETR | 118 (27 + 91) | 42.59 |
| Ours + Grounding-DINO | 187 (96 + 91) | 44.00 |
| Ours + Yolo-World | 108 (17 + 91) | 43.68 |

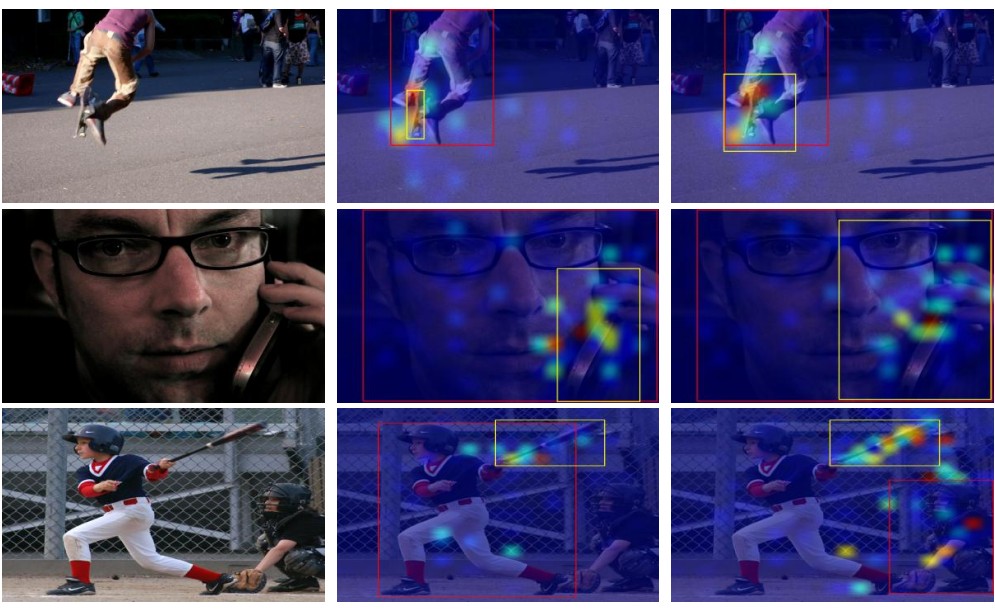

Figure 5: Visualization of cross attention map obtained from the spatial-aware pooling. The human is marked with red rectangle, while the object is marked with yellow rectangle.

interacting subject. The 4th example illustrates failure under heavy occlusion and in scenes with many visually similar objects, where unreliable detections lead to incorrect predictions.

**Visualization of Cross Attention Map in SAP.** Spatial-aware pooling can aggregate useful information from the whole image to break through the limitations of pooling features and integrate useful context information. To show how it works, we visualize the cross attention maps between the given human-object pair and the image. As shown in Fig. 5, the attention maps concentrate on the object and on relevant human body parts, such as arms and hands, that are involved in the interaction. Notably, SAP can focus on informative regions even when the detections are imperfect. In addition, the output features of SAP are also used to identify the interactiveness between humans and objects. In the 3rd example, the attention map accurately emphasizes regions crucial for deciding whether the pair is interactive.

In Fig. 6, we provide additional examples, showing that SAP adaptively focuses on different body parts for different interactions. For example, it attends to the mouth/hand and leg area when recognizing the interaction "blowing"/"riding".

### A.5 SOME EXAMPLES OF QUESTION PROMPT AND CORRESPONDING ANSWER

We show some examples of question prompt used in Table 3.

**Simple.** With the question prompt, the answer should be: sitting on an airplane, flying an airplane, riding an airplane.

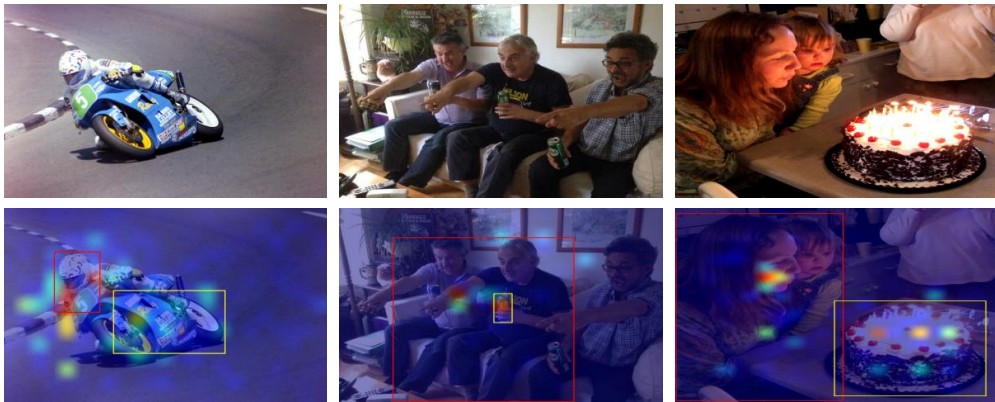

Figure 6: Visualization of more examples about cross attention map obtained from the spatial-aware pooling. The human is marked with red rectangle, while the object is marked with yellow rectangle.

---

*Question*: $<f_{\text{img}}>$ The interaction features are $<f^i_{\text{inter}}>$. Select the correct interaction from the list: boarding an airplane, directing an airplane, exiting an airplane, flying an airplane, inspecting an airplane, loading an airplane, riding an airplane, sitting on an airplane, washing an airplane.

---

**Multiple Choice Questions.** With the question prompt, the answer should be: D, G, H.

---

*Question*: $<f_{\text{img}}>$ The interaction features are $<f^i_{\text{inter}}>$. Select the correct interaction from the list: A. boarding an airplane, B. directing an airplane, C. exiting an airplane, D. flying an airplane, E. inspecting an airplane, F. loading an airplane, G. riding an airplane, H. sitting on an airplane, I. washing an airplane.

---

**In Context.** We randomly select some question-answer pair from the training set to formulate the in context examples in the question prompt. The answer should be: carrying an umbrella, holding an umbrella, standing under an umbrella.

---

I will give you some examples:
*Question*: $<f_{\text{img}}>$ The interaction features are $<f^i_{\text{inter}}>$. Select the correct interaction from the list: boarding an airplane, directing an airplane, exiting an airplane, flying an airplane, inspecting an airplane, loading an airplane, riding an airplane, sitting on an airplane, washing an airplane.
*Answer*: sitting on an airplane, flying an airplane, riding an airplane.

*Question*: $<f_{\text{img}}>$ The interaction features are $<f^i_{\text{inter}}>$. Select the correct interaction from the list: carrying a couch, lying on a couch, sitting on a couch.
*Answer*: sitting on a couch.

*Question*: $<f_{\text{img}}>$ The interaction features are $<f^i_{\text{inter}}>$. Select the correct interaction from the list: feeding a zebra, holding a zebra, petting a zebra, watching a zebra.
*Answer*: feeding a zebra, petting a zebra, watching a zebra.

According to the above examples, you should give me the answer of the question.

*Question*: $<f_{\text{img}}>$ The interaction features are $<f^i_{\text{inter}}>$. Select the correct interaction from the list: carrying an umbrella, holding an umbrella, losing an umbrella, opening an umbrella, repairing an umbrella, setting an umbrella, standing under an umbrella.

---