# OpenReview forum: "Zero-shot HOI Detection with MLLM-based Detector-agnostic Interaction Recognition"
_ICLR.cc/2026/Conference — ICLR 2026 Poster_

### Official Review · Reviewer_zR4B · 2025-10-27

**Soundness:** 3
**Presentation:** 3
**Contribution:** 3
**Rating:** 6
**Confidence:** 4

**Summary:**

This paper leverages Multi-modal Large Language Models to address HOI detection problem and decoupled object detection and interaction recognition. Interaction recognition is reformulated as a VQA task to leverage the MLLM abilities. The authors propose two pipelines, a training-free deterministic generation method, which replaces open-ended text generation with conditional likelihood estimation; and a fine-tuning pipeline, which enhances efficiency with a spatial-aware pooling module and a one-pass deterministic matching strategy. The method achieves strong zero-shot and cross-dataset performances.

**Strengths:**

1. The paper first leverages MLLM to solve HOI problem, which is novel and promising, as it connects high-level vision-language reasoning with structured HOI understanding.
2. The paper presents two pipelines, a training-free deterministic pipeline and a fine-tuned version and both achieve competitive performances.
3. Traditional MLLM-based generation is computationally expensive and unstable for classification-style tasks. The paper proposes to reformulate the open-ended generation into a deterministic generation problem. In fine-tuning version, it proposes a single-pass feature matching task using HOI tokens and cosine similarity, improving efficiency.
4. The experiments systematically compare MLLMs of different scales, demonstrating consistent improvements as model size increases (from 0.5B to 7B). This analysis provides valuable evidence that scaling multimodal models enhances HOI reasoning, further validating the proposed method.

**Weaknesses:**

1. Since the method leverages the MLLMs, naturally with strong open-vocabulary capability evaluating on open-vocabulary benchmarks such as SWiG-HOI [a] would better demonstrate its true generalization to unseen categories.
2. The paper lacks failure case analysis or qualitative discussion. Despite leveraging a large multimodal model, the best mAP remains around 45 in Table 1, 6, 7, suggesting unaddressed limitations.
3. The paper mainly improves MLLM with spatial information, while HOI detection inherently involves both spatial understanding (detection) and semantic reasoning (interaction recognition). It remains unclear whether original MLLM reasoning is already sufficient for HOI detection. A deeper discussion on this design choice (focusing on spatial information) would strengthen the overall analysis.

References:

[a] Learning Transferable Human-Object Interaction Detector with Natural Language Supervision, CVPR2022

**Questions:**

1. I have a question about the Table 2 for cross-dataset evaluation. Since V-COCO and HICO-DET have different pre-defined HOI classes, only baselines supporting to add new HOI classes in inference can be evaluated in this cross-dataset evaluation. Then, I wonder how ADA-CM is evaluated in table 2, whose fine-tuning version seems to be unable to add new classes in inference.

---

> ### Author Response · Authors · 2025-11-19
>
> ### **Q1:**
> Evaluating on open-vocabulary benchmarks such as SWiG-HOI [a] would better demonstrate the true generalization of the method to unseen categories.
>
> ### **R1:**
> Thank you for the suggestion. We have added experiments on SWiG-HOI. This challenging dataset includes over 400 actions and 1000 object categories. Our method surpasses prior work by a clear margin across all metrics, demonstrating strong open-vocabulary capability.
>
> |  Methods   | Non-rare  | Rare | Unseen | Full |
> |  ----  | ----  | ----  | ----  | ----  |
> | GEN-VLKT | 20.91 | 10.41 | - | 10.87 |
> | MP-HOI [a] | 20.28 | 14.78 | - | 12.61 |
> | CMD-SE [b] | 21.46 | 14.64 | 10.70 | 15.26 |
> | INP-CC [c] | 22.84 | 16.74 | 11.02 | 16.74 |
> |**Ours** | **25.10** | **19.65** | **15.47** | **19.86** |
>
> [a] Yang J, Li B, Zeng A, et al. Open-world human-object interaction detection via multi-modal prompts[C]//Proceedings of the ieee/cvf conference on computer vision and pattern recognition. 2024: 16954-16964.
>
> [b] Lei T, Yin S, Liu Y. Exploring the potential of large foundation models for open-vocabulary hoi detection[C]//Proceedings of the IEEE/CVF Conference on Computer Vision and Pattern Recognition. 2024: 16657-16667.
>
> [c] Lei T, Yin S, Chen Q, et al. Open-Vocabulary HOI Detection with Interaction-aware Prompt and Concept Calibration[C]//Proceedings of the IEEE/CVF International Conference on Computer Vision. 2025: 23945-23957.
>
> ### **Q2:**
> The paper lacks failure case analysis or qualitative discussion.
>
> ### **R2:**
> Thanks for the suggestion. The visualization of some failure cases is shown in Fig. 4 (Appendix). The failure of our method can be divided into three major categories:
>
> a. In many images, the model predicts correct interactions that are not annotated in the ground truth, producing false positives.
>
> b. The model incorrectly associates non-interactive human-object pairs.
>
> c. Under heavy occlusion or when visually similar objects co-occur, detection errors lead to wrong interaction recognition.
>
> We are willing to add more failure case analysis to our camera-ready version.
>
> ### **Q3:**
> It remains unclear whether original MLLM reasoning is already sufficient for HOI detection. A deeper discussion on this design choice (focusing on spatial information) would strengthen the overall analysis.
>
> ### **R3:**
> Good question. The spatial information and semantic reasoning are both important for HOI detection. We have already considered both of them in our design.
>
> To improve MLLM with spatial information, we use an object detector for object localization and propose SAP to integrate pairwise spatial information for interaction recognition.
>
> Beyond spatial understanding, our method also strengthens the MLLM’s ability to perform interaction reasoning.
> Traditional open-ended text generation of MLLMs does not model interaction semantics effectively. As shown in Table 3, it often produces hallucinated, incomplete, or invalid outputs, leading to poor IR performance. The proposed deterministic generation/matching reformulates interaction prediction as a controlled likelihood-based scoring or a features matching task, leading to a more accurate and interpretable form of interaction reasoning.
>
> Experimental results in Table 3, 4 validate the importance of the SAP and deterministic generation/matching.
>
> ### **Q4:**
> It seems only baselines supporting to add new HOI classes in inference.
>
> ### **R4:**
> Our framework naturally supports cross-dataset evaluation. The open-vocabulary detector can detect any category. For interaction recognition, the HOI classes to be recognized are defined by the candidate list in the question prompt, which can be changed freely during inference. **Therefore, evaluating on V-COCO only requires replacing the candidate list with V-COCO’s HOI classes.**
>
> ### **Q5:**
> How ADA-CM is evaluated in table 2, whose fine-tuning version seems to be unable to add new classes in inference.
>
> ### **R5:**
> Good Question. The fine-tuned version of ADA-CM cannot add new classes in inference because the Instance-Centric, Interaction-Aware, and Semantic branches are all tuned on specific class embeddings. To enable cross-dataset evaluation, **we only tune Instance-aware Adapter and re-initialize the key features with training samples in V-COCO.** This process leads to some training data linkage and may lead to a slightly higher performance of ADA-CM on V-COCO. However, even under this setting, our model outperforms ADA-CM with a clear margin.
>
> We hope these responses resolve the initial concerns. We would be grateful if you could consider raising the score to support our work. We would be glad to clarify any remaining points if you have any further questions.

---

> > ### Comment · Reviewer_zR4B · 2025-11-27
> >
> > Thanks for the author's detailed response. My original concerns are mostly well addressed.
> >
> > I also understand some concerns raised by other reviewers, particularly regarding the scope of experimental validation and the strength of the zero-shot generalization, as VLMs are considered as general models, instead of a task-specific method. I would like to see the ongoing discussion or comments from other reviewers.
> >
> > Currently, I would like to retain my positive score, as this paper firstly connects high-level vision-language reasoning with structured HOI understanding, which seems to be a good start.

---

> > > ### Author Response · Authors · 2025-11-27
> > >
> > > We are grateful that our rebuttal has resolved most of your concerns. If any questions remain, we would welcome the opportunity to discuss them further. Thank you again for guiding us toward improving our work.

---

### Official Review · Reviewer_4117 · 2025-10-27

**Soundness:** 2
**Presentation:** 3
**Contribution:** 2
**Rating:** 4
**Confidence:** 5

**Summary:**

The paper proposes a decoupled HOI pipeline that keeps object detection separate from interaction recognition and uses an MLLM to classify interactions.
Key ingredients are a “deterministic generation” proxy for multi‑label IR, a spatial‑aware pooling (SAP) module that fuses ROI features with pairwise spatial cues via cross‑attention, and a one‑pass deterministic matching scheme to score all candidate verb–object pairs in a single forward pass.
The method reports gains on HICO‑DET zero‑shot splits, a cross‑detector setting (e.g., swapping in Grounding‑DINO or YOLO‑World), and cross‑dataset transfer to V‑COCO, with extensive ablations.

**Strengths:**

- The approach yields strong zero‑shot results on HICO‑DET, including a training‑free variant (31.5 mAP), and improves further with LoRA fine‑tuning and the proposed modules; it also transfers well to V‑COCO (59.91 mAP).

- The paper demonstrates plug‑and‑play use with multiple detectors (ResNet‑50 DETR, Grounding‑DINO, YOLO‑World) without retraining the IR head. This supports the “detector‑agnostic” design goal.

**Weaknesses:**

- From both architectural and modular perspectives, the proposed method exhibits limited novelty. Architecturally, it essentially follows the conventional two-stage HOI detection paradigm, where the detection and interaction recognition stages are sequentially processed. The only modification lies in substituting these two components with more powerful counterparts, an open-vocabulary detector (e.g., Grounding-DINO) and an MLLM-based interaction recognizer, without introducing new mechanisms or architectural restructuring. At the module level, each component is constructed from widely used elements: SAP integrates ROIAlign, MLPs, and cross-attention layers, coupled with a standard pairwise spatial vector; interactiveness is estimated via a simple linear classifier; and the deterministic generation/matching merely reformulates scoring through prompt-based likelihood estimation and LoRA fine-tuning, which are established techniques in existing MLLM literature. Consequently, the methodological advancement is incremental relative to prior two-stage and CLIP/MLLM-augmented HOI frameworks

- The claimed zero‑shot setting is substantially aided by powerful pretrained components whose training corpora likely cover many HOI objects and patterns. Implementation explicitly relies on Grounding‑DINO/YOLO‑World for detection and Qwen2.5‑VL for IR, and the candidate interaction list is built from object categories, meaning much of the supervision is imported from large‑scale pretraining and the dataset’s curated verb–object space. As a result, the evaluation is closer to dataset‑level zero‑shot composition under heavy external priors than to strict zero‑data transfer.

- Although one‑pass matching and interactiveness filtering reduce the number of MLLM calls, the pipeline still couples a heavy open‑vocabulary detector with an MLLM‑based pairwise scorer, and the number of pairs scales quadratically with detections. The reported per‑image times in Table 4 appear to measure the IR component; it is unclear whether detector runtime, prompt construction, and tokenization are included, and how the overall throughput compares to strong one‑stage baselines at matched accuracy.

**Questions:**

- What exactly is included in the latency reported in Table 4? Please report end‑to‑end wall‑clock time, memory, and GPU count for detection + pairing + IR at test time, and compare to one‑stage methods at similar accuracy.

- Can the authors report results when the MLLM directly predicts HOI triplets in an open-ended manner (i.e., free-form verb–object labels per human–object pair, without the candidate list?

---

> ### Author Response · Authors · 2025-11-19
>
> ### **Q1:**
> From both architectural and modular perspectives, the proposed method exhibits limited novelty, e.g., SAP integrates ROIAlign, MLPs, and cross-attention layers, coupled with a standard pairwise spatial vector; interactiveness is estimated via a simple linear classifier; and the deterministic generation/matching merely reformulates scoring through prompt-based likelihood estimation and LoRA fine-tuning, which are established techniques in existing MLLM literature.
>
> ### **R1:**
> I would like to clarify that our core contribution lies in **introducing a decoupled HOI detection framework that leverages MLLMs for interaction recognition**. As validated in Table 1, this detector-agnostic design allows our model to benefit from advanced detectors without retraining, while simultaneously benefiting from the rich linguistic priors and reasoning capabilities of MLLMs. This detector-agnostic framework is, to our knowledge, the first in zero-shot HOI detection, and establishes a new paradigm for leveraging MLLMs in HOI detection. These contributions are also acknowledged by Reviewer zR4B and jVQb.
>
> Realizing this decoupled framework is non-trivial due to two challenges:
> (1) imperfect detections, which propagate errors to interaction recognition;
> (2) uncontrolled open-ended text generation in MLLMs, which prevents reliable scoring of candidate interactions and introduces hallucination.
> To address these challenges, we introduce two purpose-built modules: **SAP**, which integrates visual and spatial cues to make MLLMs robust against detection noise and to strengthen fine-grained interaction understanding; **Deterministic generation/matching**, which replaces open-ended text generation with constrained likelihood-based scoring, eliminating hallucinations and yielding stable interaction prediction.
>
> Ablations show that removing either module causes substantial degradation, validating that both designs are essential for enabling an effective MLLM-based decoupled HOI detector. Furthermore, **our framework is the first to simultaneously support training-free, zero-shot, cross-detector, and cross-dataset HOI detection.**
> We hope this clarification better communicates the contribution and novelty of our work.
>
> ### **Q2:**
> The claimed zero‑shot setting is substantially aided by powerful pretrained components. Implementation explicitly relies on Grounding‑DINO/YOLO‑World for detection and Qwen2.5‑VL for IR, and the candidate interaction list is built from object categories, meaning much of the supervision is imported from large‑scale pretraining and the dataset’s curated verb–object space. As a result, the evaluation is closer to dataset‑level zero‑shot composition under heavy external priors than to strict zero‑data transfer.
>
> ### **R2:**
> In this field, all methods are proposed to better mine the prior knowledge in pre-trained models, e.g., CLIP, BLIP-2, which is the key to achieving generalization to unknown categories. It is unfair to criticize our method for the integration with pre-trained models.
>
> Our zero-shot setting follows the same protocol used in prior work. Specifically, the ResNet-50 DETR is trained strictly on HICO-DET, and unseen categories are removed from the candidate list during training. Under this controlled setting, our model achieves the best performance among all methods (Table 1).
>
> **More importantly, our strong transfer performance in the cross-dataset experiments further demonstrates that the proposed framework generalizes beyond the dataset‑level zero‑shot composition.**

---

> ### Author Response · Authors · 2025-11-19
>
> ### **Q3:**
> Please report end‑to‑end wall‑clock time, memory, and GPU count for detection + pairing + IR at test time, and compare to one‑stage methods at similar accuracy.
>
> ### **R3:**
> Thanks for the suggestion. The inference is performed on a single NVIDIA 3090 GPU with a batch size of 1. We report the inference time of detection, pairing, and IR, and the GPU memory usage in the Table. The prompt construction and tokenization can be pre-computed offline. The pairing is a simple index operation. The inference time of these two parts is minor (2 *ms*).
>
> |  Methods | Detection (*ms*) | Pairing & Tokenization (*ms*) | IR (*ms*) | GPU Memory  (*GB*)|
> |  ----  | ----  | ----  | ----  | ----  |
> | Ours + ResNet50 DETR | 27 | 2 | 89 | 13.58 |
> | Ours + Grounding-DINO | 96 | 2 | 89 | 15.14 |
> | Ours + Yolo-World | 17 | 2 | 89 | 13.12 |
>
> We also compare with other methods in the Table below. The UniHOI and BC-HOI are the one-stage method. For the two-stage method, we report the detection and IR inference times.
> As shown in the table below, our method achieves competitive inference speed while delivering significantly stronger performance. Importantly, our decoupled framework enables plug-and-play detector replacement, allowing users to flexibly trade off speed and accuracy by choosing different detectors.
>
> |  Methods | Inference Time (*ms*) | *Avg* mAP |
> |  ----  | ----  | ----  |
> | UniHOI | 86 | 32.08 |
> | BC-HOI | 102 | 37.87 |
> | ADA-CM | 93 (27 + 66) | - |
> | EZ-HOI | 100 (27 + 73) | 36.20 |
> | Ours + ResNet50 DETR | 118 (27 + 91) | 42.59 |
> | Ours + Grounding-DINO | 187 (96 + 91) | 44.00 |
> | Ours + Yolo-World | 108 (17 + 91) | 43.68 |
>
> ### **Q4:**
> Can the authors report results when the MLLM directly predicts HOI triplets in an open-ended manner (i.e., free-form verb–object labels per human–object pair, without the candidate list?
>
> ### **R4:**
> Good suggestion.
> We tested the MLLM in a fully open-ended setting where it directly generates free-form HOI triplets for each human–object pair, without using any candidate list.
> **However, without the candidate list, the model has no constraints on the HOI label space and frequently predicts verbs outside the HICO-DET category set.** For example:
>
> We use the image of the 1st example in Fig. 4. The output is:
>
> *The person is using a cup.*
>
> We use the image of the 3rd example in Fig. 4. The output is:
>
> *The person is taking care a baby.*
>
> Although semantically reasonable, these verbs do not exist in the dataset’s verb vocabulary, leading to mismatches during evaluation. Under this open-ended setting, performance drops dramatically:
>
> **mAP:** 3.4%
>
> **Format error rate:** 89.15%
>
> These results confirm why deterministic generation with a candidate list is necessary.
>
> We hope these responses resolve the initial concerns. We would be grateful if you could consider raising the score to support our work. We would be glad to clarify any remaining points if you have any further questions.

---

> > ### Comment · Reviewer_4117 · 2025-11-27
> > **Response to the authors' Rebuttal**
> >
> > Thx for the reviewers' detailed rebuttal, and most of my concerns have been resolved. As the other reviewers said, this paper explores adopting VLM to handle the structural HOI detection tasks, though the two-stage framework is not a novel framework.   Thus, I decided to raise my rating to 6.

---

> > > ### Author Response · Authors · 2025-11-27
> > >
> > > We are grateful that our rebuttal has resolved most of your concerns. If any questions remain, we would welcome the opportunity to discuss them further. Thank you again for guiding us toward improving our work.

---

### Official Review · Reviewer_jVQb · 2025-10-29

**Soundness:** 3
**Presentation:** 3
**Contribution:** 3
**Rating:** 6
**Confidence:** 4

**Summary:**

This paper introduces a novel framework for zero-shot Human-Object Interaction (HOI) detection that fully decouples object detection from interaction recognition. It leverages Multi-modal Large Language Models (MLLMs) by formulating interaction recognition as a Visual Question Answering task, enabled by a deterministic generation method for training-free zero-shot inference. To further boost performance and efficiency, the authors propose a Spatial-Aware Pooling module to integrate appearance and spatial cues, and a One-Pass Deterministic Matching strategy that converts generation into efficient feature matching. Extensive experiments demonstrate state-of-the-art performance under various zero-shot settings and strong cross-dataset generalization.

**Strengths:**

- This paper investigates an interesting question of how to utilize MLLM to perform the HOI detection task.
- The presentation is clear and easy to follow.
- The proposed One-Pass Deterministic Matching is effective and interesting.

**Weaknesses:**

-  Lack of Justification for the One-Pass Prompt Design. The proposed one-pass method, which appends the same special token <|hoi|> after each candidate interaction in the prompt (Line 286), requires further theoretical and empirical justification. This design raises a fundamental concern regarding the contextual representation of the <|hoi|> tokens within the Transformer architecture. Specifically, the representation of a particular <|hoi|> token (e.g., the one following candidate T_n) is computed through self-attention over the causal context—it can attend to all preceding tokens, including the text of T_1 to T_n and all previous <|hoi|> tokens, but it is inherently blind to the subsequent candidates (T_{n+1} to T_M).
- Potential Instability Due to Candidate Order Sensitivity. Following the above question, the asymmetric contextual representation leads to a critical, unexplored question: Would the model's prediction for a given candidate interaction be sensitive to the order in which the candidates are listed? If the candidate list is permuted, the contextual information available to each <|hoi|> token changes, which may lead to inconsistent similarity scores and final predictions. Such order sensitivity would be a significant drawback for a robust system, as the candidate set is inherently unordered. The authors must empirically evaluate this potential instability, for instance, by measuring performance variance under different random permutations of the candidate list.
- Missing some efficiency comparison with the traditional HOI methods. The ablation study on inference time (Table 4) convincingly shows the efficiency gains from the proposed components. However, to better position the work, a comparison of the overall inference speed against other state-of-the-art HOI methods is necessary, such as BC-HOI and EZ-HOI. Furthermore, please specify the hardware configuration and batch size used for the timing measurements.

**Questions:**

Please kindly refer to the weakness section.

---

> ### Author Response · Authors · 2025-11-19
>
> ### **Q1:**
> Lack of Justification for the One-Pass Prompt Design.
>
> ### **R1:**
> Good Question. We have already noticed the potential impact of the interaction order in the list.
> To prevent the model from overfitting to any specific ordering of candidate interactions, we randomly shuffle the candidate list at every training iteration. This prevents the model from learning position-specific patterns and forces each <|hoi|> token to learn a local representation tied to its associated interaction rather than its absolute position.
>
> **Attention analysis.**
> We further calculate the attention weights of the <|hoi|> tokens to all candidate interactions before it. The nearest candidate interaction has about 0.94 attention weights, demonstrating that the special token learns to only aggregate information of the candidate interaction right before it.
>
> **More experiments.**
> We also expanded our experiments on alternative token placements under the Unseen-Object setting. As shown in the table, placing all <|hoi|> tokens after the interaction list leads to similar results, while placing these tokens before the list makes them blind to the correct interaction. The model cannot converge when placing all <|hoi|> tokens before the list.
>
> |  Prompts   |  Unseen | Seen | Full |
> |  ----  | ----  | ----  | ----  |
> | <\|hoi\|>T_1, <\|hoi\|>T_2, ..., <\|hoi\|>T_n | 30.18 | 25.27 | 26.08 |
> |   <\|hoi\|><\|hoi\|><\|hoi\|>,T_1, T_2, ..., T_n | - | - | - |
> |  T_1, T_2, ..., T_n, <\|hoi\|><\|hoi\|><\|hoi\|> | 48.62 | 42.55 | 43.56 |
> |T_1<\|hoi\|>, T_2<\|hoi\|>, ..., T_n<\|hoi\|> (Ours) | 48.67 | 42.58 | 43.60 |
>
> ### **Q2:**
> Potential Instability Due to Candidate Order Sensitivity.
>
> ### **R2:**
> Good Question. To prevent the model from overfitting to the order of the interactions in the list, **during training, we shuffle the candidate list at each iteration.**
> In addition, we also randomly drop some candidate interactions in the list at each iteration during training.
>
> We evaluated order robustness by performing inference 5 times using different permutations of the candidate list. The std of mAP under the Unseen-Object setting is extremely small (std ≤ 0.03), demonstrating that our model is not order sensitive.
>
> |  Unseen | Seen | Full |
> |  ----  | ----  | ----  |
> | 48.65 $\pm$ 0.03 | 42.59 $\pm$ 0.02 | 43.61 $\pm$ 0.02 |
>
> ### **Q3:**
> Missing some efficiency comparison with the traditional HOI methods.
>
> ### **R3:**
> Thank you for the suggestion. We provide a detailed inference-time comparison with other methods. All experiments are conducted on a single NVIDIA RTX 3090 GPU with a batch size of 1. For two-stage methods, we also report the detection and IR inference times.
>
> As shown in the table below, our method achieves competitive inference speed while delivering significantly stronger performance. Importantly, our decoupled framework enables plug-and-play detector replacement, allowing users to flexibly trade off speed and accuracy by choosing different detectors.
>
> |  Methods | Inference Time (*ms*) | *Avg* mAP |
> |  ----  | ----  | ----  |
> | UniHOI | 86 | 32.08 |
> | BC-HOI | 102 | 37.87 |
> | ADA-CM | 93 (27 + 66) | - |
> | EZ-HOI | 100 (27 + 73) | 36.20 |
> | Ours + ResNet50 DETR | 118 (27 + 91) | 42.59 |
> | Ours + Grounding-DINO | 187 (96 + 91) | 44.00 |
> | Ours + Yolo-World | 108 (17 + 91) | 43.68 |
>
> We hope these responses resolve the initial concerns. We would be grateful if you could consider raising the score to support our work. We would be glad to clarify any remaining points if you have any further questions.

---

> ### Comment · Reviewer_jVQb · 2025-11-27
>
> Thanks to the authors for the response. After reading the rebuttal and other reviewers' comments, I would say my concerns are largely resolved. However, I think the other reviewers' concerns are potentially valid. So my current rating remains the same, which is a positive. I would like to see if other reviewers' concerns are addressed by the rebuttal.

---

> > ### Author Response · Authors · 2025-11-27
> >
> > We are grateful that our rebuttal has resolved most of your concerns. If any questions remain, we would welcome the opportunity to discuss them further. Thank you again for guiding us toward improving our work.

---

### Official Review · Reviewer_fri8 · 2025-10-30

**Soundness:** 2
**Presentation:** 3
**Contribution:** 2
**Rating:** 4
**Confidence:** 5

**Summary:**

This paper propose a decoupled framework that separates object detection from IR and leverages multi-modal large language models (MLLMs) for training-free zero-shot IR, with a spatial-aware pooling module and a one-pass deterministic matching method to enhance performance and efficiency. Extensive experiments demonstrate that this method achieves superior zero-shot performance, strong cross-dataset generalization, and the flexibility to integrate with any object detectors without retraining.

**Strengths:**

1. Clear Motivation. The paper identifies the key limitations of existing HOI approaches, including coupling with a specific detector, poor generalization, and coarse-grained VLM features. In response, This paper propose a method that explicitly decouples object detection from interaction recognition, supports flexible plug-and-play integration with any detector, and demonstrates strong generalization capability.

2. In contrast to prior work that relies on CLIP embeddings, this paper introduces a novel deterministic generation strategy to address application bottlenecks of MLLMs in zero-shot HOI tasks. The proposed framework further implements one-pass deterministic matching, which substantially improves computational efficiency.

3. Experiments show that the proposed method surpasses other baselines under zero-shot setting, training-free setting and cross-dataset setting. Ablation studies confirm the effectiveness of each individual component in the framework.

**Weaknesses:**

1. Limited Novelty. The proposed method appears relatively simple and heavily relys on the capabilities of open-vocabulary detectors and MLLMs, which also means it inherits their inherent weaknesses, such as missed detections, redundant detections, incorrect category predictions in detectors, and hallucination issues in MLLMs.

2. The method first defines the candidate interaction list based on the categories of detected objects, thereby reformulating the HOI task into a multi-label classification problem for the MLLM. This design may considerably restrict the applicability of the model in open-world scenarios.

3. While the two datasets are widely used, they are relatively outdated(2015 & 2018). It remains unclear whether the approach can handle certain complex cases (e.g., multi-object, dense crowds situations) or perform robust open interaction recognition. It would be valuable to include experiments or visualizations demonstrating performance on new tasks mentioned in the introduction, such as robotic manipulation or autonomous driving.

4. Experimental results on the SWiG-HOI dataset are absent.

**Questions:**

See limitations and cons above.

---

> ### Author Response · Authors · 2025-11-19
>
> ### **Q1:**
> The proposed method inherits weaknesses of the detector and MLLMs, such as missed detections, redundant detections, incorrect category predictions in detectors, and hallucination issues in MLLMs.
>
> ### **R1:**
> * **About Contribution:** As agreed by Reviewer zR4B and jVQb, **our key contribution is the introduction of the decoupled framework for HOI detection with MLLMs**. Unlike prior methods (e.g., CLIP4HOI, LAIN, ADA-CM) that tightly couple interaction recognition (IR) with specific detectors, the proposed decouple framework separates detection from IR. This allows plug-and-play compatibility with any detector (Grounding-DINO, YOLO-World, etc.). To integrate both fine-grained and generalizable features for IR, MLLMs are leveraged. Our method achieves competitive performance in both training-free and fine-tuning pipelines. **This detector-agnostic paradigm is, to our knowledge, the first in zero-shot HOI detection.**
>
> * **About Incorrect Detection:** The incorrect detection is a fundamental bottleneck that affects all HOI detection methods. **It is hard for all methods to infer correct interactions from incorrect detections.**
> Object detectors have progressed far more rapidly than HOI detection models, which exhibit lower missed-detection rates, fewer redundant predictions, and more accurate category classification. **However, existing one-stage or two-stage methods cannot leverage these gains. Any detector upgrade requires retraining the model, and the IR module remains tied to the quality of the jointly trained detector.**
> In contrast, **our method can improve HOI detection performance by simply replacing the detector with a stronger one, with no additional training.** Furthermore, as shown in Table 8 (Appendix), providing ground-truth detections yields significant performance gains, confirming that the IR module is not the source of errors. These results can demonstrate that our method provides a promising way to deal with incorrect detection in HOI detection.
>
> * **About Hallucination:** Hallucination arises in open-ended text generation. Our method eliminates this failure mode by changing open-ended generation to deterministic generation/matching. **This restricts the MLLM to select from candidate interaction list rather than generating free-form text.**
> As shown in Table 3, without our deterministic generation, format error and single-output rates are high. **With our design, all such errors drop to zero, demonstrating that hallucination is avoided.**
>
> ### **Q2:**
> Defining the candidate interaction list based on the categories of detected objects may considerably restrict the applicability of the model in open-world scenarios.
>
> ### **R2:**
> I would like to clarify that the candidate interaction list can be changed freely during inference. Therefore, the model can recognize any HOI category and is not restricted to a fixed interaction vocabulary.
> For example, if the candidate list is:
>
> * [feeding a bird, watching a bird, touching a bird], the model recognizes all three;
>
> if the list is:
>
> * [watching a bird, touching a bird], the model recognizes two.
>
> **This is similar to the open-vocabulary HOI detection setting that defines interactions through text prompts.**
>
> We restrict the unseen interactions during training.
> In zero-shot and cross-dataset settings, unseen interactions are excluded during training and added back into the candidate list at inference. As shown in Tables 1 and 2, our method achieves strong performance on these unseen interactions, demonstrating its open-world generalization capabilities.

---

> ### Author Response · Authors · 2025-11-19
>
> ### **Q3:**
> While the HICO-DET and V-COCO are widely used, they are relatively outdated(2015 & 2018). It remains unclear whether the approach can handle certain complex cases (e.g., multi-object, dense crowds situations) or perform robust open interaction recognition. It would be valuable to include experiments or visualizations demonstrating performance on new tasks mentioned in the introduction, such as robotic manipulation or autonomous driving.
>
> ### **R3:**
> Thanks for the suggestion. First, as detailed in **R4**, we evaluate our method on the large-scale SWIG-HOI benchmark, which contains over 400 actions and 1000 object categories, including many multi-object and cluttered scenes. Our method surpasses previous state-of-the-art methods by a substantial margin, highlighting its strong open-vocabulary generalization capabilities.
>
> Second, to evaluate performance on tasks that more closely align with real-world embodied AI scenarios, we additionally conduct experiments on the ego-centric hand–object interaction dataset Enigma-51 [a]. Given an input image, Enigma-51 requires predicting the triplet $<$ hand, hand contact state, active object $>$, which is a core capability for robotic manipulation in embodied AI. To test the transferability of our method, we directly apply the model trained on HICO-DET without any retraining. We provide ground-truth hand and object bounding boxes as detection results and simply replace the candidate interaction list with the verb set of Enigma-51 to perform hand contact state recognition. Our method achieves 72.49% mAP. Even without training on this dataset, the performance of our method is comparable with the method trained with the in-domain data [a] (73.31\%), demonstrating strong generalization to ego-centric viewpoints and fine-grained contact states even without training on this dataset. Moreover, the ablation study shows that the proposed SAP and deterministic matching are also effective on the ego-centric hand-object interaction detection task.
>
> |  Settings   | Results (mAP) |
> |  ----  | ----  |
> | w/o SAP | 71.33 |
> | w/o DM | 70.21 |
> |**Ours** | **72.49** |
>
> [a] Ragusa F, Leonardi R, Mazzamuto M, et al. Enigma-51: Towards a fine-grained understanding of human behavior in industrial scenarios[C]//Proceedings of the IEEE/CVF Winter Conference on Applications of Computer Vision. 2024: 4549-4559.
>
> ### **Q4:**
> Experimental results on the SWiG-HOI dataset are absent.
>
> ### **R4:**
> Thank you for the suggestion. We have added experiments on SWiG-HOI. This challenging dataset includes over 400 actions and 1000 object categories. Our method surpasses prior work by a clear margin across all metrics, demonstrating strong open-vocabulary capability.
>
> |  Methods   | Non-rare  | Rare | Unseen | Full |
> |  ----  | ----  | ----  | ----  | ----  |
> | GEN-VLKT | 20.91 | 10.41 | - | 10.87 |
> | MP-HOI [b] | 20.28 | 14.78 | - | 12.61 |
> | CMD-SE [c] | 21.46 | 14.64 | 10.70 | 15.26 |
> | INP-CC [d] | 22.84 | 16.74 | 11.02 | 16.74 |
> |**Ours** | **25.10** | **19.65** | **15.47** | **19.86** |
>
> [b] Yang J, Li B, Zeng A, et al. Open-world human-object interaction detection via multi-modal prompts[C]//Proceedings of the ieee/cvf conference on computer vision and pattern recognition. 2024: 16954-16964.
>
> [c] Lei T, Yin S, Liu Y. Exploring the potential of large foundation models for open-vocabulary hoi detection[C]//Proceedings of the IEEE/CVF Conference on Computer Vision and Pattern Recognition. 2024: 16657-16667.
>
> [d] Lei T, Yin S, Chen Q, et al. Open-Vocabulary HOI Detection with Interaction-aware Prompt and Concept Calibration[C]//Proceedings of the IEEE/CVF International Conference on Computer Vision. 2025: 23945-23957.
>
> We hope these responses resolve the initial concerns. We would be grateful if you could consider raising the score to support our work. We would be glad to clarify any remaining points if you have any further questions.

---

> > ### Comment · Reviewer_fri8 · 2025-11-27
> > **Official Comment by Reviewer fri8**
> >
> > Thanks to the authors for the detailed response. While the "decoupled framework" proposed in this paper is highly dependent on a specific MLLM, which contributes to the decoupling of the two sub-networks. Additionally, since this work frees up the object decoder, it indeed relies on a sophisticated MLLM. There is also a lack of comprehensive comparisons with MLLM-based HOI detection baselines. Taking all these factors into consideration, I will retain my initial rating for now and look forward to feedback from the other reviewers.

---

> > > ### Author Response · Authors · 2025-11-27
> > >
> > > Thank you for your thoughtful feedback. We appreciate your comments and would like to further address each of your concerns in detail:
> > >
> > > ### **Q1:**
> > > The "decoupled framework" proposed in this paper is highly dependent on a specific MLLM.
> > >
> > > ### **R1:**
> > > We would like to clarify that our proposed decoupled framework is designed to be flexible and can be extended to various MLLMs. As shown in Table 6, our method is effective with different MLLMs, e.g., LLaVA Onevision and Qwen2.5-VL. Additionally, we show that the framework is scalable across MLLMs of different sizes
> > >
> > > ### **Q2:**
> > > This work frees up the object decoder, it indeed relies on a sophisticated MLLM.
> > >
> > > ### **R2:**
> > > The integration of MLLMs plays a crucial role in the effectiveness of our decoupled framework. The interaction recognition requires fine-grained and generalized features, which MLLMs are well-equipped to provide. In contrast, existing methods rely on coarse-grained CLIP features. These methods need to introduce additional detector features to enhance the CLIP features with instance-level representation capability.
> > >
> > > To highlight the importance of MLLMs in our framework, **we conducted an experiment using only CLIP features for interaction recognition. The mAP in this setting is only 31.7%.** This result further validates the need for MLLMs in decoupling object detection and interaction recognition.
> > > In addition, the increased computational complexity of our method is also acceptable compared with existing methods.
> > >
> > > **The use of MLLMs for various vision tasks is an emerging trend in the community.** Our work follows this trend, and we believe our contribution of integrating MLLMs into HOI detection is significant, as also acknowledged by reviewers zR4B and jVQb.
> > >
> > > ### **Q3:**
> > > A lack of comprehensive comparisons with MLLM-based HOI detection baselines.
> > >
> > > ### **R3:**
> > > As shown in Table 4, integrating MLLMs for HOI detection is not trivial. Our proposed SAP and Deterministic Generation/Matching effectively address challenges such as the imperfect detections and uncontrolled open-ended text generation issues, which are key to the final results.
> > >
> > > To further investigate the role of MLLMs in HOI detection, **we conducted an experiment to assess whether the MLLM alone can handle both detection and interaction reasoning.** Specifically, we fine-tuned the MLLM to jointly predict human-object pairs and interaction labels. The format of the training data is as follows:
> > >
> > > *Question:* $<$ $f_{\text{img}}$ $>$ Detect all the human-object pairs and recognize their interaction. The candidate interactions are: [feeding a bird, holding a bird, riding a bike, sitting on a bike, ......].
> > >
> > > *Answer:*
> > >
> > > Human [0, 100, 200, 150], Bird [50, 120, 98, 150], feeding, holding a bird.
> > >
> > > Human [0, 100, 200, 150], Bike [20, 50, 348, 130], riding a bike.
> > >
> > > After training, the mAP under a fully supervised setting is only 22.16%.
> > > This significant drop in performance is primarily due to the poor object detection capability of the MLLM, which aligns with findings from other work [a].
> > > Therefore, decoupling object detection from IR and using the MLLM only for IR is a reasonable design.
> > >
> > > [a] Li J, Xie C, Ao J, et al. LMM-Det: Make Large Multimodal Models Excel in Object Detection[C]//Proceedings of the IEEE/CVF International Conference on Computer Vision. 2025: 308-318.
> > >
> > > We hope that these clarifications address the concerns raised by the reviewer. We are glad to clarify any remaining points if you have any further questions. It is really important to us if you could consider raising the score to support our work.

---

### Author Response · Authors · 2025-11-19

We thank the reviewers for their constructive comments and acknowledgement of our contributions. These are really helpful and inspiring. All reviewers agreed that our motivation is clear and the paper is easy to follow (Reviewer fri8, jVQb, 4117, zR4B). Our proposed framework that decouples object detection and interaction recognition in HOI detection is interesting (Reviewer jVQb, 4117, zR4B), and also establishes a new paradigm for leveraging MLLMs to perform interaction recognition (Reviewer jVQb, zR4B). The flexible plug-and-play integration with any detector is interesting and promising (Reviewer fri8, 4117). Reviewers also acknowledge our promising performance under training-free, zero-shot, cross-dataset, and cross-detector settings (Reviewer fri8, jVQb, 4117, zR4B). We would like to revise our paper after discussion with reviewers. The code will be released.

We hope the substantially improved results on SWiG-HOI, strong cross-dataset and cross-domain transfer (including to robotic manipulation), and the demonstrations of stability, efficiency, and design justification fully resolve the concerns. We remain grateful for the reviewers’ efforts and hope that the new evidence helps them reassess the contribution and significance of the work in a positive direction. We would be glad to clarify any remaining points.

# Common Concerns
## 1. The experimental results of SWIG-HOI (Reviewer fri8, zR4B).
Thanks for the suggestion. We have added experiments on SWiG-HOI. This challenging dataset includes over 400 actions and 1000 object categories. Our method surpasses prior work by a clear margin across all metrics, demonstrating strong open-vocabulary capability.

|  Methods   | Non-rare  | Rare | Unseen | Full |
|  ----  | ----  | ----  | ----  | ----  |
| GEN-VLKT | 20.91 | 10.41 | - | 10.87 |
| MP-HOI [a] | 20.28 | 14.78 | - | 12.61 |
| CMD-SE [b] | 21.46 | 14.64 | 10.70 | 15.26 |
| INP-CC [c] | 22.84 | 16.74 | 11.02 | 16.74 |
|**Ours** | **25.10** | **19.65** | **15.47** | **19.86** |

## 2. The inference time (Reviewer jVQb, 4117).
Thanks for the suggestion. We provide a detailed inference-time comparison with other methods. All experiments are conducted on a single NVIDIA RTX 3090 GPU with a batch size of 1. For two-stage methods, we also report the detection and IR inference times.

As shown in the table below, our method achieves competitive inference speed while delivering significantly stronger performance. Importantly, our decoupled framework enables plug-and-play detector replacement, allowing users to flexibly trade off speed and accuracy by choosing an appropriate detector.

|  Methods | Inference Time (*ms*) | *Avg* mAP |
|  ----  | ----  | ----  |
| UniHOI | 86 | 32.08 |
| BC-HOI | 102 | 37.87 |
| ADA-CM | 93 (27 + 66) | - |
| EZ-HOI | 100 (27 + 73) | 36.20 |
| Ours + ResNet50 DETR | 118 (27 + 91) | 42.59 |
| Ours + Grounding-DINO | 187 (96 + 91) | 44.00 |
| Ours + Yolo-World | 108 (17 + 91) | 43.68 |

## 3. The contribution (Reviewer fri8, 4117).
I would like to clarify that our core contribution lies in **introducing a decoupled HOI detection framework that leverages MLLMs for interaction recognition**. As validated in Table 1, this detector-agnostic design allows our model to benefit from advanced detectors without retraining, while simultaneously benefiting from the rich linguistic priors and reasoning capabilities of MLLMs. This detector-agnostic framework is, to our knowledge, the first in zero-shot HOI detection, and establishes a new paradigm for leveraging MLLMs in HOI detection. These contributions are also acknowledged by Reviewer zR4B and jVQb.

Realizing this decoupled framework is non-trivial due to two challenges:
(1) imperfect detections, which propagate errors to interaction recognition;
(2) uncontrolled open-ended text generation in MLLMs, which prevents reliable scoring of candidate interactions and introduces hallucination.
To address these challenges, we introduce two purpose-built modules: **SAP**, which integrates visual and spatial cues to make MLLMs robust against detection noise and to strengthen fine-grained interaction understanding; **Deterministic generation/matching**, which replaces open-ended text generation with constrained likelihood-based scoring, eliminating hallucinations and yielding stable interaction prediction.

Ablations show that removing either module causes substantial degradation, validating that both designs are essential for enabling an effective MLLM-based decoupled HOI detection model. Furthermore, **our framework is the first to simultaneously support training-free, zero-shot, cross-detector, and cross-dataset HOI detection.**
We hope this clarification better communicates the contribution and novelty of our work.

---

### Comment · Area_Chair_q43W · 2025-11-25
**Rebuttal Review Request**

Dear Reviewers,

The authors have responded to your reviews. Please engage in the discussion and evaluate the authors’ rebuttal to determine whether your comments have been adequately addressed.

Best, Your AC

---

### Author Response · Authors · 2025-11-26
**Invitation for Further Discussion**

We sincerely appreciate the reviewer’s time and effort in evaluating our submission. We would be very grateful to receive your feedback. If there are any remaining concerns, we are fully open to discussion and would be happy to provide more details or conduct supplementary analyses. Thank you again for your valuable insights and constructive comments.

---

### Author Response · Authors · 2025-12-01
**Summary of rebuttal**

Dear Program Chairs, Senior Area Chairs, Area Chairs, and Reviewers:

We understand the significant challenges posed by the recent OpenReview incident and sincerely appreciate your efforts in continuing the review process. Because the review scores have been reverted to their pre-rebuttal state, we would like to provide a factual summary of the constructive exchanges we had with the reviewers.

Before the system reset, our rebuttal had successfully addressed key reviewer concerns and led to a score improvement from **(6, 6, 4, 4) to (6, 6, 6, 4)**. To assist in your assessment, we provide a timeline of interactions and a summary of the concerns that were addressed.

## 1. Summary of Strengths
**Decouple framework:** All reviewers agree that the proposed framework that decouples object detection and interaction recognition in HOI detection is interesting and different from existing methods.

**Leveraging MLLMs for HOI detection:** Reviewers 4117, jVQb, zR4B acknowledge that our method establishes a new paradigm for leveraging MLLMs to perform interaction recognition.

**Plug-and-play integration of detector:** The flexible plug-and-play integration with any detector is interesting and promising (Reviewers fri8, 4117).

**Promising performance:** All reviewers acknowledge that our method is the first to simultaneously support training-free, zero-shot, cross-detector, and cross-dataset HOI detection.

## 2. Timeline of Events
We highlight the timeline below to show that improvements in the review scores were the direct result of discussion, and that the positive acknowledgements preceded widespread awareness of the OpenReview data leak. We were busy responding to additional concerns posted by Reviewer fri8 during the OpenReview data leak.

**Nov 11:** Rebuttal period started.

**Nov 19:** We submitted detailed responses.

**Nov 27 13:26:** Reviewer jVQb acknowledged that our response addressed his concerns and maintained the initial positive score (6 -> 6).

**Nov 27 15:30:** Reviewer zR4B acknowledged that his concerns were mostly well addressed and maintained the initial positive score (6 -> 6).

**Nov 27 16:14:** Reviewer fri8 posted additional concerns about the effectiveness of our method on other MLLMs, and the lack of comprehensive comparisons with MLLM-based HOI detection baselines.

**Nov 27 23:47:** We submitted further responses and new experiments addressing Reviewer fri8.

**Nov 27 23:53:** Reviewer 4117 acknowledged that most of his concerns were resolved and raised his score (4 -> 6).

**Nov 28:** ICLR/OpenReview officially disclosed the data leak and announced the score reversion.

---

> ### Author Response · Authors · 2025-12-01
>
> ## 3. Summary of Resolved Concerns
> ### Reviewer fri8 (4)
>
> * **The proposed method inherits weaknesses of the detector and MLLMs:** We clarified that our method can improve HOI detection performance by simply replacing the detector with a stronger one, with no additional training. The proposed deterministic generation/matching addresses the hallucination of MLLMs.
>
> * **The applicability of the model in open-world scenarios:** We demonstrated the open-world generalization capabilities of our method.
>
> * **The experiments on new tasks should be included:** We added the experiments on the ego-centric hand–object interaction dataset Enigma-51 and validated the effectiveness of our method.
>
> * **Experimental results on the SWiG-HOI dataset:** We added the experiments on SWIG-HOI and demonstrated the promising performance of our method.
>
> After the first round, this reviewer posted additional concerns, which we addressed in detail:
>
> * **The "decoupled framework" proposed in this paper is highly dependent on a specific MLLM:** Table 6 demonstrates scalability across different and differently sized MLLMs (LLaVA Onevision, Qwen2.5-VL).
>
> * **The method relies on a sophisticated MLLM:** We clarified that the MLLMs is important to our fully decoupled framework. Leveraging the MLLMs for vision tasks is also an emerging trend in the community.
>
> * **A lack of comprehensive comparisons with MLLM-based HOI detection baselines:** We added these experiments and validated that decoupling object detection from IR and using the MLLM only for IR is a reasonable design.
>
> We believe that our response can address the concerns of reviewer fri8.
>
> ### Reviewer jVQb (6 -> 6)
> * **Candidate order sensitivity and one-pass prompt design:** We clarified the training settings of our method to prevent the model from overfitting to any specific ordering of candidate interactions. Some extra experiments were conducted to validate that our model is not order sensitive.
>
> * **Efficiency analysis:** We provided an inference time comparison. Our method shows competitive efficiency.
>
> ### Reviewer 4117 (4 -> 6)
> * **Limited novelty:** We clarified our core contribution of the decoupled framework for HOI detection, which is the first to simultaneously support training-free, zero-shot, cross-detector, and cross-dataset HOI detection.
>
> * **Zero-shot generalization capability:** We clarified that our method follows the same setting as the existing method and shows a stronger zero-shot generalization capability.
>
> * **Efficiency analysis:** We provided an inference time comparison. Our method shows competitive efficiency.
>
> * **Report results when the MLLM directly predicts HOI triplets in an open-ended manner:** We added these experiments and validated the necessity of the proposed deterministic generation.
>
> ### Reviewer zR4B (6 -> 6)
> * **Experimental results on the SWiG-HOI dataset:** We added the experiments on SWIG-HOI and demonstrated the promising performance of our method.
>
> * **The paper lacks failure case analysis:** We clarified that the failure case analysis is included in Fig. 4.
>
> * **A deeper discussion on the design choice (focusing on spatial information) should be included:** We clarified that our method has already considered both spatial information and semantic reasoning in our design.
>
> * **Only baselines supporting adding new HOI classes in inference:** We clarified that our method can add new HOI classes by replacing the candidate list.
>
> * **The cross-dataset evaluation of ADA-CM:** We provided clarification on its integration into V-COCO.
>
> A positive consensus was reached before or only shortly after the official announcement of the leak. We hope this summary clarifies the current state of the manuscript and the reviewers’ satisfaction with our revisions before the system reset.
>
> Thank you for your time and fair consideration.
>
> Sincerely,
>
> Authors

---

### Meta-Review · Area_Chair_C9NS · 2026-01-04

**Summary:**

Reviewers generally find the detector-agnostic decoupling (detector for boxes + MLLM for interaction recognition) and the deterministic generation/matching idea promising, with strong reported zero-shot/cross-detector/cross-dataset results. The remaining deciding concern is whether the contribution is sufficiently novel and convincingly validated, especially given the reliance on strong pretrained detectors/MLLMs and questions about broader comparisons and practicality (end-to-end cost). The AC learns to accept.

**Reviewer Concerns:**

Addressed by rebuttal: added SWiG-HOI and Enigma-51 results; provided inference-time breakdown (detection + pairing/tokenization + IR) and comparisons vs HOI baselines; analyzed candidate-order sensitivity and prompt design (shuffle training, permutation std); clarified why deterministic scoring is needed vs open-ended generation; added discussion/failure modes and additional ablations.

Still outstanding: one reviewer has concerns on novelty (pipeline looks like a stronger two-stage stack) and concerns about dependence on specific MLLMs / completeness of MLLM-based HOI baseline comparisons may remain partially open, though authors added extra MLLM-scaling and comparisons.

**Reviewer Scores:**

- fri8 (4): likely 4 → 4, raised new concerns in rebuttal
- jVQb (6): 6 → 6, explicitly says concerns largely resolved; keeps positive score.
- 4117 (4): 4 → 6, concerns resolved, explicitly increased after rebuttal.
- zR4B (6): 6 → 6, explicitly retains positive score; concerns mostly addressed.

---

### Decision · Program_Chairs · 2026-01-26

Accept (Poster)